# Specific Antimicrobial Activities Revealed by Comparative Evaluation of Selected Gemmotherapy Extracts

**DOI:** 10.3390/antibiotics13020181

**Published:** 2024-02-13

**Authors:** Melinda Héjja, Emőke Mihok, Amina Alaya, Maria Jolji, Éva György, Noemi Meszaros, Violeta Turcus, Neli Kinga Oláh, Endre Máthé

**Affiliations:** 1Doctoral School of Nutrition and Food Science, Faculty of Agricultural and Food Sciences and Environmental Management, University of Debrecen, Böszörményi Str. 128, 4032 Debrecen, Hungary; hejja.melinda@agr.unideb.hu (M.H.); maria.jolji@agr.unideb.hu (M.J.); 2Institute of Nutrition Science, Faculty of Agricultural and Food Sciences and Environmental Management, University of Debrecen, Böszörményi Str. 128, 4032 Debrecen, Hungary; mihokemoke@uni.sapientia.ro (E.M.); aleya.amina@agr.unideb.hu (A.A.); 3Doctoral School of Animal Science, Faculty of Agricultural and Food Sciences and Environmental Management, University of Debrecen, Böszörményi Str. 128, 4032 Debrecen, Hungary; 4Department of Food Science, Faculty of Economics, Socio-Human Sciences and Engineering, Sapientia Hungarian University of Transylvania, Libertății sq. 1., 530104 Miercurea Ciuc, Romania; gyorgyeva@uni.sapientia.ro; 5Department of life Sciences, Faculty of Medicine, Vasile Goldis Western University of Arad, L. Rebreanu Str. 86, 310414 Arad, Romania; noe.meszaros@gmail.com (N.M.); violeta_buruiana@yahoo.com (V.T.); 6CE-MONT Mountain Economy Center, Costin C. Kirițescu National Institute of Economic Research, Romanian Academy, Petreni Str. 49, 725700 Suceava, Romania; 7Department of Pharmaceutical Chemistry, Faculty of Pharmacy, Vasile Goldis, Western University of Arad, L. Rebreanu Str. 86, 310414 Arad, Romania; neli.olah@plantextrakt.ro; 8PlantExtrakt Ltd., No. 46, 407059 Cluj, Romania

**Keywords:** gemmotherapy extracts, antimicrobial activity, antifungal activity, *Morus nigra*, *Juglans regia*, *Prunus amygdalus*, *Olea europaea*, *Ribes nigrum*, *Rubus fruticosus*, *Vaccinium myrtillus*

## Abstract

Nowadays, unprecedented health challenges are urging novel solutions to address antimicrobial resistance as multidrug-resistant strains of bacteria, yeasts and moulds are emerging. Such microorganisms can cause food and feed spoilage, food poisoning and even more severe diseases, resulting in human death. In order to overcome this phenomenon, it is essential to identify novel antimicrobials that are naturally occurring, biologically effective and increasingly safe for human use. The development of gemmotherapy extracts (GTEs) using plant parts such as buds and young shoots has emerged as a novel approach to treat/prevent human conditions due to their associated antidiabetic, anti-inflammatory and/or antimicrobial properties that all require careful evaluations. Seven GTEs obtained from plant species like the olive (*Olea europaea* L.), almond (*Prunus amygdalus* L.), black mulberry (*Morus nigra* L.), walnut (*Juglans regia* L.), blackberry (*Rubus fruticosus* L.), blackcurrant (*Ribes nigrum* L.) and bilberry (*Vaccinium myrtillus* L.) were tested for their antimicrobial efficiency via agar diffusion and microbroth dilution methods. The antimicrobial activity was assessed for eight bacterial (*Bacillus cereus*, *Staphylococcus aureus*, *Salmonella enterica* subsp. *enterica*, *Proteus vulgaris*, *Enterococcus faecalis*, *Escherichia coli*, *Pseudomonas aeruginosa* and *Listeria monocytogenes*), five moulds (*Aspergillus flavus*, *Aspergillus niger*, *Aspergillus ochraceus*, *Penicillium citrinum*, *Penicillium expansum*) and one yeast strain (*Saccharomyces cerevisiae*). The agar diffusion method revealed the blackberry GTE as the most effective since it inhibited the growth of three bacterial, four moulds and one yeast species, having considered the total number of affected microorganism species. Next to the blackberry, the olive GTE appeared to be the second most efficient, suppressing five bacterial strains but no moulds or yeasts. The minimum inhibitory concentration (MIC) and minimum bactericidal concentration (MBC) were then determined for each GTE and the microorganisms tested. Noticeably, the olive GTE appeared to feature the strongest bacteriostatic and bactericidal outcome, displaying specificity for *S. aureus*, *E. faecalis* and *L. monocytogenes*. The other GTEs, such as blueberry, walnut, black mulberry and almond (the list indicates relative strength), were more effective at suppressing microbial growth than inducing microbial death. However, some species specificities were also evident, while the blackcurrant GTE had no significant antimicrobial activity. Having seen the antimicrobial properties of the analysed GTEs, especially the olive and black mulberry GTEs, these could be envisioned as potential antimicrobials that might enhance antibiotic therapies efficiency, while the blackberry GTE would act as an antifungal agent. Some of the GTE mixtures analysed have shown interesting antimicrobial synergies, and all the antimicrobial effects observed argue for extending these studies to include pathological microorganisms.

## 1. Introduction

Many studies claim that antimicrobial resistance (AMR) is one of the major public health problems and that, if nothing is done, AMR could be one of the leading causes of mortality by 2050 [1]. To this day, an estimated 700,000 people die yearly from AMR-related diseases, and without an effective plan, an estimated 10 million people could die from the same causes by the year 2050. This is not only a scientifically challenging task but also bears an enormous economic cost, by some estimates at around USD 100 trillion of worth [2]. One of the main reasons for this happening is the liberal use of antibiotics, as well as the relatively limited number of available types of antibiotics, leading to an ever-increasing number of drug-resistant processes in bacteria [3]. It is a relatively well-known fact that pathogenic bacteria adapt to different antibacterial agents, and these can be divided into four major categories: drug uptake limitation, drug target modification, drug inactivation and active drug efflux. Drug resistance processes can also vary based on the type of bacteria (Gram-negative or Gram-positive), changing their structures. Out of the thirty-two types of antibacterial agents in development in 2019, only six were considered novel by the WHO, and, as such, the development of a very good quality antimicrobial drug remains a major problem. Recently, there has been an increasing interest in natural substance-based antibiotics. While the existing types of antimicrobial agents remain in use, albeit with reduced effectiveness, and there are some new types of synthetically manufactured drugs, one can argue that natural substances show great promise in this regard [2]. 

These natural substances also showed potential in preventing diseases caused by drug-tolerant and resistant strains of microorganisms. There are several studies that prove that plant extracts have antimicrobial effects against multidrug-resistant bacteria. For example, Mascarello et al. (2018) reported that eight compounds isolated from mulberry root barks were capable of *M. tuberculosis* protein tyrosine phosphatase inhibition, a bacterium that increased the number of cases in multidrug-resistant tuberculosis [4]. Walnut bark and leaf extracts were effective against methicillin-resistant *Staphylococcus aureus* [5,6], *Salmonella enterica* serovar Typhi, *Enterobacter cloacae* [5] and *Pseudomonas aeruginosa* [5,7]. Ildiz et al. (2021) reported that bilberry fruit extracts have antimicrobial effects against multidrug-resistant *Escherichia coli* and *Pseudomonas aeruginosa* [8]. There are approximately 374,000 plant species in the world [9] that produce many phytochemicals as secondary metabolites to protect themselves against biotic and abiotic stresses [10]. It has been suggested that the most important categories of chemical compounds found in plants with proven antimicrobial activity are terpenoids, alkaloids, sulphur-containing compounds and polyphenols. Not only do they provide defence against various microorganisms, but also show anti-fungal, anticancer and antioxidant properties as well [11]. It is assumed that the structural arrangement of the different compounds found in these plants influences the effectiveness of the antimicrobial property; one such structure is the hydroxyl (-OH) group of a phenol compound. It is a known fact that this group can directly disrupt a microbial membrane structure and cause leakages [12]. A number of documented chemical compounds, such as flavonoids, quinones, tannins, lignans and other polyphenols, are also effective as antifungal agents [13]. These compounds are also potentially effective in food preservation, as they are not only considered nutritionally safe and easily digestible but also have proven health benefits [14,15].

Most of the research focuses on the extracts made from differentiated tissues, but in recent years, there have been more and more studies based on meristematic tissues. This type of tissue contains many phytonutrients, and they are used for the preparation of so-called gemmotherapy extracts (GTEs) that are more complex and potentially more effective than adult plant parts [16]. Besides the secondary metabolites, these GTEs usually contain proteins, amino acids, hormones, vitamins, growth factors and cytokines [17]. The best time to harvest the buds is late winter or early spring, as this is when the maximum concentration of active ingredients is present [18]. In reviewing the studies that have been carried out in this area, we conclude that there is only a small amount of research that focuses on the efficacy of the antimicrobial properties of such GTEs, so this area is largely unexplored. The application of GTEs in phytotherapy has been proposed by Pol Henry [19], and such extracts are hydroglycerine alcoholic solutions of macerated fresh buds, stems, roots, or other meristematic plant tissues, and are easy to prepare and administer, usually by dilution in water [20,21]. In the following, some of the recent results are presented concerning the studied GTEs associated antimicrobial properties.

### 1.1. Olive Tree (Olea europaea L.)

The olive (*Olea europaea* L.) belongs to the *Oleaceae* family, and it is a largely studied Mediterranean crop because of its health benefits, such as its antioxidant, antimicrobial, anticancer, antiviral and gastroprotective effects [22]. Young olive shoot extracts’ antimicrobial effects were first examined in 2023 by Popović et al. [16], and according to them these extracts have inhibitory and bactericidal effects against several Gram-positive (*Staphylococcus aureus* ATCC 25923, *Bacillus cereus* ATCC 14579, *Listeria monocytogenes* ATCC 14579, *Enterococcus faecalis* ATCC 29212) and Gram-negative (*Escherichia coli* ATCC 25922, *Salmonella enteritidis* ATCC 13076) bacteria. Recent research proved that olive leaf extract might have a protective effect against the negative impacts of exposure to noise and toluene on the heart tissue, so it could be a safe and natural therapeutic option to counteract the adverse impact of environmental toxicants on cardiovascular health [23]. Another work of research claimed that olive leaf extract has wound-healing effects and can increase hair growth by detoxifying the hair follicles and promoting blood circulation in the scalp [24]. 

### 1.2. Almond (Prunus amygdalus L.)

The almond (*Prunus amygdalus* L.) belongs to the *Rosaceae* family, and it is cultivated in east Mediterranean countries, Australia and Africa [25]. The skin of the nut has anti-inflammatory and antioxidant activities due to the presence of nutrients, such as lipids, fatty acids, proteins, amino acids, vitamins, carbohydrates and minerals [26]. The sphingolipids isolated from nuts are capable of inhibiting the development of colon cancer and also decreasing the proportion of adenocarcinomas in mice [25]. There are no studies based on the antimicrobial effects of buds, but Ibibia (2013) [25] reported that almond leaf extracts were effective against *E. coli*, *S. aureus*, *B. subtilis*, *B. cereus* and *P. aeruginosa.* Ramachandran et al. (2020) [27] reported that almonds could be a promising agent for the treatment of polycystic ovary syndrome, as it was able to restore hormone levels in experimental animals.

### 1.3. Black Mulberry (Morus nigra L.)

The black mulberry (*Morus nigra* L.) belongs to the *Moraceae* family, and there are 24 species of *Morus* and one subspecies. The black mulberry originates from Iran, and it is one of the most important species grown in Mediterranean countries. The fruit is known not only for its nutritional qualities but also for its use in traditional medicine, as it contains a high number of bioactive compounds. The fruit is reported to have several biological activities, such as antioxidant, antidiabetic, anti-hyperlipidaemic and anti-inflammatory properties [28]. The leaves can be used for the prevention of throat infections, inflammations and irritations, and it is an anti-hyperglycemic nutraceutical food for diabetic people. The root barks have cathartic and anti-helmentic effects, while the stem barks are considered to be purgative and antidiabetic [29].

### 1.4. Walnut (Juglans regia L.)

The walnut (*Juglans regia* L.) belongs to the *Juglandaceae* family, and it originates from Central Asia, the western Himalayan chain, reaching Europe before the Roman times [30]. The walnut is rich in fats and contains valuable polyunsaturated fatty acids, proteins and minerals. Walnut-derived polyphenolic compounds have been attributed to health benefits like the ability to reduce oxidative stress and inhibit macromolecular oxidation. Regular walnut consumption reduces the risk of heart disease [31]. Oliveira et al. (2008) [32] examined the walnut’s green husk antimicrobial properties, observing the growth inhibition of *B. cereus*, *B. subtilis*, *S. aureus* and *P. aeruginosa*, while walnut fruits were less effective, as reported by Pereira et al. (2008), [33]. Another study on walnut kernel extracts reported increased brain dopamine levels in rats, which were inversely correlated with oxidative stress, suggesting that such an extract could have neuroprotective effects [34].

### 1.5. Bilberry (Vaccinium myrtillus L.)

The bilberry (*Vaccinium myrtillus* L.) belongs to the *Ericaceae* family and is a wild-growing species with its most important distribution area in Northern and Eastern Europe and North Africa. These berries have been used in traditional medicine since ancient times as they have antioxidant, cardioprotective, antimicrobial, anti-inflammatory, anti-obesity and other beneficial health properties [35]. There are many studies that examine these berries and other parts of the plant for associated antimicrobial properties [36,37,38,39], but no such study was conducted for the bilberry GTE. However, some bilberry fruit extracts’ specific phytonutrient profiles were determined and the antidiabetic effect was revealed through nutrigenetic studies (Neamțu et al., 2020) [40]. Habanova et al. (2016) [41] reported that even a short period of regular consumption of whole wild bilberries is associated with an improvement in the lipid profile in humans and can contribute to beneficial effects on CVD risk reduction, such as decreasing LDL-C and TG and increasing HDL-C.

### 1.6. Blackberry (Rubus fruticosus L.) 

The blackberry (*Rubus fruticosus* L.) belongs to the *Rosaceae* family, and it has been collected for about two thousand years from the wild, though currently, many cultivated varieties are also available. Nowadays, the main regions for blackberry production are North America, Europe, Asia, South America, Central America and Africa [42]. The berries were used for medicinal purposes, and the plants were domesticated in the 17th century [43]. Due to their phenolic compounds, these berries have been attributed to many potential health benefits, such as the prevention of chronic and inflammatory diseases and some types of cancer, and could also reduce the changes associated with age-related neurodegenerative diseases [42].

### 1.7. Blackcurrant (Ribes nigrum L.)

The blackcurrant (*Ribes nigrum* L.) belongs to the *Glossulariaceae* family, and it is native to central Europe and northern Asia. It is also a well-studied plant; its berries have many positive effects on health, like blood glucose regulation, as well as anti-inflammatory, antioxidant, antimicrobial and anticancer properties [44]. Raiciu et al. (2010) [18] investigated the antimicrobial effects of blackcurrant bud extracts and found that they inhibited the growth of *Staphylococcus aureus* ATCC6538, *Pseudomonas aeruginosa* ATCC 9027, *Escherichia coli* ATCC 35218, *Aspergillus niger* ATCC 16,404 and *Candida albicans* ATCC 10231. Tabart et al. (2006; 2011) [45,46] compared the antioxidant capacities and profiles of different plant parts (buds, leaves and berries) from different blackcurrant cultivars, and the berries exhibited a significantly diminished phenol content compared to the buds and leaves. Kendir et al. (2019) [47] reported that blackcurrant leaf extracts had wound-healing effects, the activity of which could be attributed to phenolic compounds, especially chlorogenic acid and rutin. Another work of research showed that blackcurrant GTEs prevented neuroinflammation in rats that underwent lipopolysaccharide treatment [48].

### 1.8. Aim of the Research

Seven GTEs made of plant species like *Olea europaea* (OGTE), *Prunus amygdalus* (AGTE), *Morus nigra* (BMGTE), *Juglans regia* (WGTE), *Rubus fruticosus* (BkBGTE), *Ribes nigrum* (BCGTE) and *Vaccinium myrtillus* (BBGTE) were comparatively assessed for their antimicrobial properties by monitoring the growth and function of food-derived pathogens (*Escherichia coli*, *Pseudomonas aeruginosa*, *Salmonella enterica* subsp. *enterica*, *Proteus vulgaris*, *Bacillus cereus*, *Staphylococcus aureus*, *Enterococcus faecalis*, *Aspergillus flavus*, *A. niger*, *A. ochraceus*, *Penicillium citrinum*, *P. expansum*) and *Saccharomyces cerevisiae*. The ultimate goal was to determine which of the tested GTEs could feature antimicrobial properties and if such properties could further be influenced by combining GTEs through their synergistic, additive and/or antagonistic interactions. Therefore, if an antimicrobial interaction is detected between some GTEs, then their possible combination for other putative beneficial physiological effects should be further investigated.

## 2. Results

In order to gain information about the antimicrobial activity of the selected GTE Gram-positive and negative bacterial species, together with yeast and mould fungi species, were included in this study (see Section 4). 

All studied GTEs were evaluated for their polyphenol contents in order to have a better characterisation of them previous to the study. The identified and quantified bioactive compounds from the studied GTEs are presented in Table 1. 

The presence of many polyphenols can be observed both from the phenolic acid class and flavonoids. Generally, the main compounds identified are caffeic and chlorogenic acids, respectively, and quercetin and its derivatives hyperoside and rutoside. Certainly, each GTE has its own specificity. The OGTE contains a high amount of luteolin-*7-O*-glucoside; in the meantime, the WGTE has more flavonoids and fewer phenolic acids. In the BkBGTE, we found a higher amount of salicylic acid and apigenin. 

We used two types of experimental setups to assess the putative antimicrobial activity of the selected GTEs. The agar diffusion method was applied to test the extent of the selected GTE-specific antimicrobial effect. This was based on the ability of the tested GTE to diffuse in the agar medium, forming a circular zone where its concentration decreased from the centre to the periphery. The resulting concentration gradient of the putative antimicrobial GTE provides an indication of the sensitivity of a given microorganism as a function of complete or partial inhibition. 

In addition to the agar diffusion method, we also analysed the antimicrobial activity in broth by means of dilutions (microbroth dilution assay). Broths containing the tested GTEs at various concentrations were inoculated with a defined amount of microbial suspension, and after incubation, the dilutions that inhibited the microorganism being tested were determined. As the microbial cells in this liquid culture media came into direct contact with the nutrient broth containing a given concentration of the antimicrobial GTE, an accurate result was obtained regarding the change in cell number (decrease, complete inhibition–destruction). Therefore, based on the broth culture, we determined the minimum inhibitory concentration (MIC) as the lowest concentration of a given GTE with antimicrobial properties that still cause a decrease in the cell number of the tested microbe. We also evaluated the minimum bactericidal concentration (MBC), which is the lowest concentration of an antimicrobial GTE that destroys 99.9–100% of the original cell’s suspension. The MIC test indicates the lowest level of antimicrobial agent that strongly inhibits growth, while the MBC test indicates the lowest level of antimicrobial agent that results in microbial death.

### 2.1. GTEs’ Antimicrobial Activity in the Context of Inhibition Zones

The antibacterial testing of selected GTEs using the agar diffusion method was continued whenever possible by evaluating the diameter of the inhibition zones. The larger the diameter of the inhibition zone, the more sensitive the microorganism tested and the lower the amount required to inhibit the microorganism (see Figure 1). 

Figure 1 The inhibition zones generated through the agar diffusion method for some of the tested microorganisms and GTEs.

Data on the inhibition zone diameters produced by the GTEs evaluated are presented in Table 2. Among the Gram-positive bacteria, *B. cereus* was the most sensitive, with the exception of blackberry and blackcurrant GTEs that showed no inhibitory effects.

The walnut GTE excelled particularly, inhibiting the growth of some microbes, even at a concentration of 20%, as the most effective against *B. cereus*, *L. monocytogenes*, and *E. faecalis*. For *S. aureus*, only the blackberry and olive GTEs produced zones of inhibition. Remarkably, for *E. faecalis*, the almond GTE looked like the most effective, followed by the blueberry and walnut GTEs. 

In the case of *L. monocytogenes*, the walnut and blackberry GTEs were the most relevant, showing inhibitory effects at a wide range of 100–20% of the GTE’s concentration. Interestingly, *P. vulgaris* was the only Gram-negative bacteria on which most GTEs had some effect, being mostly sensitive to the bilberry GTE, inhibiting growth even at a 30% concentration, and behaving completely resistant to blackcurrant GTE, where even the fully concentrated extract failed to inhibit growth.

In the case of the budding yeast, the almond GTE produced significant zones of inhibition, followed by the blackberry, blueberry and blackcurrant GTEs. Despite the fact that for moulds, no significant zones of inhibition were observed, and even colonies were found within the inhibition ring, the blackberry GTEs seemed to be an exception as they showed some inhibitory effect.

### 2.2. GTEs’ Antimicrobial Activity as Revealed by the Agar Diffusion Method

The application of this method in the case of the GTEs under investigation showed variable antimicrobial activity. The blackberry (*Rubus fruticosus*) GTE has been shown to be effective for eight strains, the olive (*Olea europaea*) GTE and blueberry (*Vaccinium myrtillus* L.) GTE for five strains, the almond (*Prunus amygdalus* L.) GTE and walnut (*Juglans regia* L.) GTE for four, and the black mulberry (*Morus nigra* L.) GTE and blackcurrant (*Ribes nigrum* L.) GTE for only two microbial species each (see Table 3). Notably, the olive GTE was effective at inhibiting Gram-negative bacteria, albeit to a variable extent (50–100% concentration), and the blackberry GTE showed inhibitory effects against mould species such as *Aspergillus flavus* (*A. flavus*), *Aspergillus ochraceus* (*A. ochraceus*), *Penicillium citrinum* (*P. citrinum*) and *Penicillium expansum* (*P. expansum*), although at a high 70% concentration. 

Remarkably, the most effective GTE based on the number of inhibited growths proved to be the blackberry (*Rubus fruticosus*), showing bacterial and fungal growth inhibition for eight out of the fourteen examined strains. One of the most sensitive microorganisms to the blackberry GTE proved to be the *Listeria monocytogenes* (*L. monocytogenes*), where even a 20% (*v*/*v*) extract concentration exhibited growth inhibition. In the case of *Staphylococcus aureus* (*S. aureus*), a concentration of 40% showed growth inhibition, while in the case of *Proteus vulgaris* (*P. vulgaris*), only a concentration of 80% proved effective. *Bacillus cereus* (*B. cereus*), *Pseudomonas aeruginosa* (*P. aeruginosa*), *Escherichia coli* (*E. coli*) and *Salmonella enterica* (*S. enterica*) proved to be totally resistant to the blackberry GTE. Based on the results for the microscopic fungi, all except the *Aspergillus niger* (*A. niger*) strain were sensitive to the blackberry GTE. 

The second most potent antimicrobial effect was found for the GTEs of olive (*Olea europaea* L.) and bilberry (*Vaccinium myrtillus* L.). The olive GTE inhibited the growth of 5 strains out of the examined 14. It can be noted that a concentration of 50% olive GTE was effective against *B. cereus*, *S. aureus* and *P. vulgaris*, while the concentrated GTE (100%) showed inhibition against *Enterococcus faecalis* (*E. faecalis*) and *Listeria monocytogenes* (*L. monocytogenes*). Unfortunately, it did not show any effectiveness for the rest of the Gram-negative strains, where not even the concentrated solutions showed any growth inhibition. Moreover, the olive GTE did not affect the growth of budding yeast. The bilberry (*Vaccinium myrtillus* L.) GTE inhibited the growth of five strains of microorganisms, such as *B. cereus*, *E. faecalis* and *P. vulgaris*, and was also effective against *Saccharomyces cerevisiae* (*S. cerevisiae*) and *Aspergillus flavus* (*A. flavus*). In the case of the bilberry GTE, for the tested bacteria, the effective minimum concentration varied between 30 and 70%, while for the moulds and yeast, it was 70%. 

Next in line are the almond (*Prunus amygdalus* L.) and walnut (*Juglans regia* L.) GTEs, which proved effective against four tested microorganism species. The almond extract showed inhibition for *B. cereus*, *E. faecalis* and *P. vulgaris*, but also for *S. cerevisiae*. The almond GTE was effective against *E. faecalis* even at a 30% dilution, against *P. vulgaris* at 40% and against *B. cereus* at 70%. The minimum effective concentration in the case of *S. cerevisiae* was 60%. The walnut GTE was efficient against *B. cereus*, *E. faecalis*, *L. monocytogenes* and *P. vulgaris*, starting from a concentration of 20% all the way up to 100%. The walnut GTE did not show effectiveness for any of the tested fungi.

The least effective GTEs were the black mulberry (*Morus nigra* L.) and blackcurrant (*Ribes nigrum* L.), which showed efficacy against only two types of microorganisms currently assessed. The black mulberry GTE inhibited the growth of *B. cereus* at a 70% concentration and *P. vulgaris* at 80%. In the case of the blackcurrant GTE, the lowest concentration was 70% for the *S. cerevisiae*, and it showed effectiveness at a 90% concentration for *L. monocytogenes*. 

Finally, it is important to mention that in our research, using the agar diffusion method, none of the studied GTEs were found to inhibit the growth of *Escherichia coli* (*E. coli*), *Salmonella enterica* subsp. *enterica* (*S. enterica* subsp. *enterica*), *Pseudomonas aeruginosa* (*P. aeruginosa*) and *Aspergillus niger* (*A. niger*). Taking together the olive, almond, black mulberry, walnut and blackcurrant GTEs, they did not feature any antimicrobial effects on the tested mould species, as revealed using the agar diffusion assay. 

### 2.3. Revealing the Antimicrobial Efficacy of Different GTE Mixtures

As described in the Materials and Methods section, different mixtures of GTEs were prepared, and their antimicrobial activity was tested using the agar diffusion method, and the diameter of the inhibition zones was quantified. The results of these tests are presented in Table 4. 

The results exhibited three types of outcomes for the mixed GTEs and their antimicrobial effectiveness based on the presence and/or the diameter of the inhibition zone generated by the tested bacterial strains. It can be reported that most of the mixed GTEs performed worse than their single-use variants, resulting in smaller inhibition zone diameters.

For *B. cereus* with a 1:1 mixture of olive and walnut GTEs, the inhibition diameter was 10.07 ± 0.61 mm, which is almost identical with 9.68 ± 0.6 mm of 50% olive GTE and 10.22 ± 0.52 mm, corresponding to the 50% walnut GTE, respectively. The olive and almond GTE mixture had an inhibition zone of 11.13 ± 0.36 mm, which is similar to the 9.68 ± 0.6 mm of the 50% olive GTE but significantly increased compared to the non-existent antimicrobial effect of the 50% almond GTE, suggesting that the almond GTE did not have any inhibitory effect on the olive GTE. Similarly, the walnut and bilberry GTE mixture induced an inhibition zone of 12.1 ± 0.66 mm that resembled 10.22 ± 0.52 mm corresponding to the 50% walnut GTE, while the 50% bilberry GTE had no detectable antimicrobial effect. The latest situation is that the blueberry GTE was not able to inhibit the antimicrobial property of the olive GTE. Furthermore, the mixture of blackcurrant and almond GTEs produced an inhibition zone of 9.16 ± 0.22 mm compared to the ineffective antimicrobial activity of the individual 50% blackcurrant and 50% almond GTEs, which is an example of a synergistic type of interaction between the above-mentioned GTEs. A similar situation was observed in the case of the mixture of blueberry and almond GTEs, where an inhibitory zone of 9.48 ± 0.29 mm was observed, despite the fact that the individual 50% GTEs did not produce any antimicrobial effect, which, again, suggests a synergistic interaction between the blueberry and almond GTEs. Another such synergistic interaction was found for the GTE combination of the bilberry and black mulberry and the GTE mixture of the black mulberry combined with almond. 

Further to the *B. cereus* and in the case of *S. aureus*, synergistic interactions were found for GTE mixtures like the bilberry–almond and walnut–black mulberry combinations. The olive–almond GTE mixture induced a 16.91 ± 1.02 mm diameter that exceeded significantly the 50% olive GTE, which generated a 9.24 ± 0.15 mm diameter, while the 50% almond GTE had no inhibition zone. Seeing the extent of the olive–almond GTE mixture-associated effect, another synergistic interaction could be envisioned between the olive and almond GTEs. Again, the walnut and blackcurrant GTE mixtures could be envisioned as synergies. The olive–blackberry GTE mixture showed neither synergy nor additive effects but a basic antimicrobial effect, while the size of the inhibition zone was near that of the olive GTE. The bilberry–olive, blackberry–black mulberry and walnut–blackberry GTE mixtures showed the basic type of antimicrobial effects that were close to one of the GTE components.

One of the most interesting GTE mixtures proved to be the walnut and blackcurrant combination for *S. aureus* bacteria. Neither of these extracts showed efficacy alone, but when mixed together, their synergy resulted in an inhibition zone of 10.87 ± 0.58 mm. The same synergy could also be envisioned for the GTE mix of almond and bilberry, both of which have no effect on *S. aureus* on their own, but when mixed together, they form a zone of inhibition of 16.57 ± 0.27 mm. 

### 2.4. Assessing GTE-Associated Minimal Inhibitory Concentrations—MIC Test

In order to further characterise the antimicrobial potential of the GTEs, we analysed the minimum inhibitory concentration (MIC), which represents the lowest concentration of a given GTE that still causes a decrease in the cell number of the tested microbe by inhibiting bacterial growth. The results of the MIC assay are presented in Table 5, where the seven evaluated GTEs and their associated MIC values are shown in concordance with the microorganisms tested.

The MIC assay is based on the colour change in the compound resazurin to resorufin [49] due to the metabolic activities of the bacterium and is a valuable non-invasive method for measuring cell numbers, as seen in Figure 2.

In the case of the MIC assay, each GTE displayed an inhibitory effect for all the tested microorganisms at different concentrations.

In the case of the olive GTE, even a 10% extract concentration showed an inhibitory effect on the *B. cereus* microbe, while for *S. aureus* and *E. faecalis*, the MIC value reached 20%. The most resistant were *E. coli* and *S. cerevisiae*; in their cases, only olive GTE concentrations above 50% inhibited reproduction. 

For the almond, two bacteria, *B. cereus* and *P. vulgaris*, proved to be significantly sensitive to the 10% GTE, while *S. cerevisiae* (90%) proved to be the most resistant in this test. 

It can be seen that all other Gram-positive bacteria were inhibited at medium concentrations, such as 40%, while Gram-negative bacteria showed a wider range of growth inhibition based on concentrations between 10% and 70%. 

The 10% GTE concentration of the black mulberry already inhibited the *B. cereus*, while the 20% concentration was effective in the case of *S. aureus* and *P. vulgaris* bacteria. The highest concentration needed for Gram-positive bacteria was measured at 50% for *E. faecalis*, while in the case of Gram-negatives, the highest value was also 50% for *E. coli* and *S. enterica*. *S. cerevisiae* yeast proved to be more sensitive as a 40% concentration was sufficient to inhibit growth. 

The walnut GTE also inhibited the microbe *B. cereus* the most at a low concentration of 20%, but *S. aureus*, *L. monocytogenes* and *P. vulgaris* also proved to be sensitive, and in their case, the inhibitions started from the 30% extract. The most resistant were *E. coli* and *S. enterica*, at 50 and 60%, while yeast also proved resistant at 70%.

In the case of bilberry GTE, *B. cereus* and *P. vulgaris* were the most sensitive since inhibition was observed at a 20% concentration. All Gram-positive bacteria were easily inhibited at a lower concentration of 30%. The Gram-negative bacteria showed higher resistance to this extract, requiring 40–50% concentrations for inhibition. *S. cerevisiae* proved to be the most resistant to this extract, requiring a 60% concentration for growth inhibition. 

The blackberry GTE showed significant growth inhibition in the case of *P. vulgaris* (Gram-negative bacteria) and *B. cereus* (Gram-positive bacteria), requiring as low as a 10% and 20% GTE concentration for growth inhibition, respectively. The Gram-positive bacteria proved significantly more resistant to this extract, requiring 60% concentration for supressing growth. The remaining Gram-negative bacteria showed varied results, but still required higher concentrations, between 40 and 60%, to inhibit growth. The most resistant microbes were *S. aureus*, *L. monocytogenes* and *S. cerevisiae*. 

The blackcurrant GTE showed mixed results in the MIC test, with the highest inhibition of *P. vulgaris* at 20% but no effect on *E. faecalis*. All the Gram-positive bacteria showed a high level of resistance to the extract at 70%. Gram-negative bacteria were inhibited, but also at higher concentrations, with *S. enterica* requiring a 100% concentrated GTE for inhibition. As usual, the yeast required higher concentrations for inhibition, which, in this case, was 80% GTE. 

In summary, the GTEs showed bacteriostatic activity against several types of microorganisms capable of causing major human infections, including *S. aureus*, *L. monocytogenes*, *E. coli* and *S. enterica*, as inferred from the observed MIC values.

### 2.5. Analysis of GTE-Associated Minimal Bactericidal Concentrations—MBC Assay

The MBC test indicates the lowest level of an antimicrobial agent, which, in our case, is the GTE concentration that results in microbial death. The results of the MBC test are presented in Table 6. The analysed GTEs showed a different MBC based on their concentrations and the microorganism species assessed. 

The olive (*Olea europaea*) and bilberry (*Vaccinium myrtillus* L.) GTEs showed an increased efficacy against almost identical six strains of both bacteria and yeast. Both types of GTE showed bactericidal activity against the same Gram-positive bacteria, with the exception of *Bacillus cereus*. In the case of olive GTE, the MBC with the lowest concentration was around 50% for *E. faecalis*, followed by 60% of *S. aureus* and 70% for *L. monocytogenes*. The bilberry GTE with the lowest MBC was 30% for *E. faecalis*, followed by 60% of *L. monocytogenes* and 80% for *S. aureus*. Furthermore, when the Gram-negative bacteria were analysed, both GTEs showed some effectiveness, but at much higher concentrations (70–80%). The current study revealed both *E. coli* and *S. enterica* presented an MBC value of 70% olive GTE, while the bilberry GTE had no bactericidal effect on *E. coli*. In addition, the bilberry GTE demonstrated an MBC of 70% for *P. aeruginosa* and an 80% GTE for *S. enterica*. It should be noticed that both olive and bilberry GTEs proved effective against *S. cerevisiae* but at very high concentrations of 90–100%. 

In the Materials and Methods chapter, we proposed a test for verifying the authenticity of the MBC colour-changing method, and Figure 3 presents the obtained results for such a test. This figure also shows the concentration (%) used for the GTE and the initial and final number of colony-forming units.

In the case of the olive GTE, the inhibitory effects on *S. enterica* species were observed at the 30% GTE concentration, where bacteria were still able to survive, while at higher concentrations in the 60–70% range, bacterial growth inhibition was more pronounced. At even higher concentrations, like 80–100%, the olive GTE was completely bactericidal. Overall, this experiment clearly demonstrated the validity of olive GTE-specific MBC data.

The second most effective GTE was the black mulberry (*Morus nigra* L.), killing five types of microorganisms at various concentrations, starting from concentrations of 50% to 100%. The black mulberry GTE showed bactericidal effects for most of the Gram-positive bacteria, except for *B. cereus.* Among the Gram-negative bacteria, only *S. enterica* subsp. *enterica* exhibited a 100% MBC, while the yeast *S. cerevisiae* again displayed a 100% concentration-specific MBC.

The third most effective GTE for the MBC test was the blackberry (*Rubus fruticosus* L.), which demonstrated effective bactericidal power against four types of microorganisms. The blackberry GTE proved not to be effective against *B. cereus* but was efficacious against the remainder of Gram-positive bacteria, even at a concentration as low as 40% (*L. monocytogenes*). It also proved effective against the yeast *S. cerevisiae*, but only at a concentration of 100%.

The fourth most effective GTEs are almond (*Prunus amygdalus* L.) and walnut (*Ju-glans regia* L.) extracts, which showed bactericidal activity against only three types of bacterial microorganisms. The almond extract completely stopped the growth of two Gram-positive bacteria, *E. faecalis* and *L. monocytogenes*, but proved ineffective against *B. cereus* and *S. aureus*. In the case of almond GTE and Gram-negative bacteria, it was only efficacious against *P. aeruginosa* and only at a 100% concentration. The walnut GTE also proved effective against two Gram-positive bacteria, *E. faecalis* and *L. monocytogenes*, at concentrations of 90% and 80%, but showed ineffective against *B. cereus* and *S. aureus*. Among the Gram-negative strains, the walnut GTE presented an MBC only against *S. enterica* and at a 100% concentration. However, neither the almond nor the walnut GTEs featured any effectiveness for the yeast *S. cerevisiae*. 

In terms of the MBC, the least effective GTE was blackcurrant (*Ribes nigrum* L.), as it had no bactericidal effect for any of the concentrations tested.

## 3. Discussion

In recent years, interest in natural antimicrobials has increased, and more studies are being carried out with a focus on plant extracts [50]. However, there is a paucity of data in the literature on the antimicrobial activity of plant parts, such as buds and young shoots, despite the fact that they contain many bioactive compounds and could have potent antimicrobial properties. In addition, recent studies have described the phytonutrient profile of some GTEs while also revealing their nutritional, antidiabetic and anti-inflammatory effects [40,48,51]. The phytonutrient composition of these GTEs also suggests that they may have antimicrobial properties, and this article presents relevant research data. 

The phytochemical characterisation of GTEs showed that they are rich in polyphenols. Polyphenols are well known for their antioxidant activity, but they also have several other biological effects, including antimicrobial activity. Studies have shown that quercetin destroys the cell wall of bacteria, alters cell permeability, modifies protein synthesis and expression, reduces the activity of bacterial enzymes and inhibits nucleic acid synthesis, appearing effective against *Pseudomonas aeruginosa, Salmonella enteritidis, Staphylococcus aureus, Escherichia coli*, *Proteus* and *Aspergillus flavus* [52]. Their derivatives release quercetin in the body, which has the same effect on bacteria. Our study and the results presented in Table 1 show that hyperoside and rutoside, two of the best-known derivatives of quercetin, are the most represented flavonoids in all the GTEs studied.

Other studies have demonstrated that caffeic acid and its derivatives, alongside chlorogenic acid, could potentiate the effect of antibiotics and, in this way, be more effective in the fight against some types of resistant pathogenic bacteria like *Staphylococcus aureus* [53]. Our results show that the GTEs studied are rich in chlorogenic acid (see Table 1). However, besides the assessed polyphenols, there could be other bioactive constituents responsible for the detected antimicrobial activity. 

### 3.1. Olive GTE Emerges as a Potent Antimicrobial against B. cereus, S. aureus and E. faecalis

As shown by the agar diffusion method, our data indicate that olive GTE inhibited the growth of *B. cereus*, *S. aureus*, *E. faecalis*, *L. monocytogenes* and *P. vulgaris* (see Table 2). The 100% concentrated olive GTE induced for *B. cereus* an inhibition zone of 12.34 ±0.46 mm, in the case of *S. aureus*, a zone of 10.52 ± 0.18 mm, for *E. faecalis*, 10.95 ± 0.5 mm, for *L. monocytogenes*, 9.71 ± 0.62 mm and for *P. vulgaris*, 11.09 ± 0.33 mm. These data suggest a certain concentration dependency that does not feature regular proportionality (see Table 2 and Figure 4). Furthermore, similar inhibition zone sizes were measured for the 50% olive GTE concentration in the case of *B. cereus*, *S. aureus* and *P. vulgaris*, for the 60% concentration in the case of *P. vulgaris* and for the 100% concentration in the case of *L. monocytogenes*. These data suggest that different olive GTE concentrations have similar inhibitory effects on different microbes. The extent of the inhibitory effect may depend on the synergetic interaction of certain olive GTE-specific phytonutrients, and the nature of this interaction is not necessarily concentration-dependent. 

Indeed, Aleya and colab. (2023) reported that olive GTE contains flavonoids, iridoids and polyphenols, which might confer some antimicrobial effects [51]. Aliabadi and colab. (2012) investigated the antimicrobial effect of Iranian olive leaf extracts and found that different concentrations of the extract have an inhibitory effect on *B. cereus*, *E. coli*, *Klebsiella pneumonia*, *Salmonella typhimurium* and *S. aureus* bacteria [54]. It is remarkable that the above-mentioned observations for the olive leaf extracts are in agreement with our findings for the olive GTE. Himour and colab. (2017) examined Algerian olive leaves and, in their case, the extract inhibited the bacteria *S. aureus*, *E. coli*, *Klebsiella pneumonia* and *P. aeruginosa* [55]. Nora and colab. (2012) tested the aqueous Algerian olive leaf extracts, and their effectiveness was proven against microbes like *E. coli* ATCC25922, *P. aeruginosa* ATCC10145, *Klebsiella pneumoniae*, *Enterobacter cloacae* ATCC13047, *S. aureus* ATCC6538 and ATCC25923 and *Bacillus stearothermophilus* ATCC11778 [56]. Moreover, Borges and colab. (2020) investigated the antimicrobial effect of Portuguese olive leaf extracts obtained through different methods, and in their case, the ultrasound-assisted ethanolic extract best inhibited the growth of *S. aureus* and *E. coli* microbes [57]. Furthermore, Gökmen and colab. (2014) investigated the effect of a Turkish olive leaf extract, and in their case the extract was effective against *B. cereus* (12.34 ± 0.46), *S. aureus* (18.67 ± 1.53), *Enterococcus faecalis* (19 ± 1.73), *Listeria monocytogenes* (19.33 ± 0.58), *Proteus vulgaris* (17.33 ± 1.53), *E. coli* (18 ± 1), *E. coli* O157 (17.67 ± 0.58), *Salmonella typhimurium* (13.33 ± 2.08), *Enterobacter sakazakii* (18.33 ± 1.15) and *P. aeruginosa* (18 ± 1.73), [58]. Taken together, these studies based on olive leaves indicate that the corresponding extracts contain oleuropein, hydroxytyrosol, chlorogenic acid, caffeic acid, verbascoside and rutin polyphenols [59], all of which could contribute to antimicrobial efficacy [60,61]. It is also interesting that caffeic acid was not identified in the olive GTE [51]. It is, therefore, logical that, based on the specificity of the phytonutrient profiles of the olive leaves extract and GTE, some differences are foreseen with regard to their antimicrobial activities.

In order to gain more information on the bacteriostatic and bactericidal activity of the analysed GTEs, we determined the associated MIC and MBC values (see Section 4). The MIC shows the lowest level of antimicrobial substance that strongly inhibits growth (bacteriostatic effect), while the MBC indicates the lowest level of the antimicrobial agent that results in microbial death (bactericidal effect). In the case of the olive GTE, the MIC values are relatively reduced since even the lowest concentration at 10% could exert a bacteriostatic effect (see Table 5 and Table 7). Interestingly, there was some evidence of a bacteriostatic effect on all the microorganisms tested, with *B. cereus* being the most sensitive and *E. coli*, together with *S. cerevisiae* being the most resistant. Regarding the MBC, in the case of *E. faecalis*, the 50% extract concentration induced bacterial death, while no bactericidal effect was detected in the case of *B. cereus*, *P. vulgaris* and *P. aeruginosa*. In the research conducted by Sudjana and colab. (2009), the MIC and MBC concentration of Australian olive leaf extracts ranged from 12.5% to 50% for *Bacillus cereus*, *E. faecalis*, *E. coli*, *L. monocytogenes*, *P. aeruginosa* and *S. enterica* [62]. In another study, the MIC value of a Turkish olive leaf extract was 32 mg/mL for *B. cereus*, *S. aureus*, *E. faecalis*, *P. vulgaris* and *E. coli*, while *L. monocytogenes*, *E. coli* O157 and *P. aeruginosa* showed 64 mg/mL [58]. The above MIC and MBC values and those reported by others do not appear to differ significantly from our data, suggesting that olive GTE and leaf extracts have similar antimicrobial activities.

### 3.2. Blackberry GTE Is Effective against L. monocytogenes, S. aureus and P. vulgaris

The blackberry (*Rubus fruticosus* L.) GTE was found to be effective, with inhibition observed in eight of the fourteen microorganisms, as shown by the agar diffusion method (see Table 2 and Table 7). The most sensitive microbe proved to be *L. monocytogenes*; the 100% concentrated blackberry GTE resulted in an inhibition zone of 19.20 ± 0.87 mm (Figure 1c) and even the 20% extract generated an inhibitory diameter of 13.4 ± 0.53 mm. In addition, significant zones of inhibition were visible for *S. aureus*, and a less pronounced inhibition was observed for *P. vulgaris* and *S. cerevisiae*. Moreover, no real inhibition was observed in the case of moulds, as colonies appeared in the inhibition zones produced, but it should be stressed that the GTE was able to produce some kind of inhibition, which represents a newly discovered effect. Weli and colab. (2020) investigated the antimicrobial effect of the Omani blackberry leaf extracts and found that the extract exerted an inhibitory effect on *E. coli*, *Haemophilus influenza*, *E. faecalis* and *S. aureus* bacteria [63]. Riaz and colab. (2011) examined the different plant parts of Pakistani blackberry, and based on their results, 100 µg of the stem extract as disks were effective against *E. coli*, *S. typhi*, *S. aureus*, *Proteus mirabilis*, *Micrococcus luteus*, *Citrobacter*, *B. subtilis* and *P. aeruginosa* bacteria [64]. Pavlović and colab. (2016) investigated the phenolic composition of different Serbian *Rubus* species and found that ellagic acid was the main phenolic acid in the leaf extract, and the largest amount was found in the blackberry leaf extract [65]. Also present in the leaf were the hydroxycinnamic acids, aesculin, catechin, myricetin, rutin, quercetin and kaempferol, the constituents of which can contribute to antimicrobial effects. It should also be noted that the presence of the above-mentioned phytonutrients was also confirmed in our HPLC-ESI-MS study. Some novel compounds, such as amino acids (4-hydroxyisoleucine and tryptophan), were also confirmed.

In the broth dilution analysis, the bacteriostatic effect of blackberry GTE was observed. *P. vulgaris* proved to be the most sensitive (MIC = 10%), while *S. aureus*, *E. faecalis*, *S. enterica* and *S. cerevisiae* were the least sensitive (MIC = 60%) (see Table 5 and Table 7). Furthermore, when the MBC was analysed, it was observed that *L. monocytogenes* was the most efficient (40%), while *S. aureus* and *E. faecalis* showed a bactericidal effect at *60%* and *70%* GTE concentrations (see Table 6). The MBC value specific to *S. cerevisiae* was equivalent to 100% of the extract. Gil-Martínez and colab. (2023) investigated the antimicrobial effect of Spanish blackberry fruit and found that the MBC was 25 mg/mL for *L. monocytogenes* and *S. aureus*, 12.5 mg/mL for *E. faecalis*, *B. cereus*, *E. coli* and *S enterica*, and 100 mg/mL for *P. aeruginosa* [66]. Concerning the bactericidal effect and comparing their data with ours, it seems reasonable to predict that the extract from the blackberry fruit appears to be more effective than our GTE.

### 3.3. The Bilberry GTE Is a Relatively Effective Antimicrobial Agent against P. vulgaris, E. faecalis and L. monocytogenes

The agar diffusion method revealed the bilberry (*Vaccinium myrtillus* L.) GTE is the most effective in the case of *P. vulgaris* bacteria, as the 100% extract produced an inhibition zone of 12.76 ± 0.80 mm (Figure 1b), and then all the decreasing concentrations reaching even the 30% extract had inhibitory effects (10.15 ± 0.45 mm inhibition diameter), (see Table 2 and Table 7). 

For the other microbes, inhibition zones were detected for *B. cereus*, *E. faecalis* and *S. cerevisiae*. Based on the data in Table 2, there was no significant difference (*p* = 0.182) between the inhibition zones produced by the 90–50% GTE concentrations in the case of *E. faecalis* bacteria, meaning that these dilutions had an identical effect. In young bilberry shoots and leaves, including the GTE, the main phenolic classes are hydroxycinnamic acids, flavonols and proanthocyanidines, meaning that similar antimicrobial effects are expected to emerge [40,67]. 

The most sensitive bacteria to the bilberry GTE in the broth MIC assay method were *B. cereus* and *P. vulgaris* with a 10% extract concentration, which was shortly followed by *S. aureus*, *E. faecalis* and *L. monocytogenes* at the 20% concentration. The *S. cerevisiae* proved to be the least sensitive (see Table 5 and Table 7). The MBC-generated bactericidal effect was detectable at a 30% GTE concentration for *E. faecalis*, whereas *L. monocytogenes* showed a 60% concentration value. Miljković and colab. (2018) investigated the antimicrobial properties of Serbian bilberry fruit, and their methanolic extract was effective against a significant number of Gram-positive and Gram-negative microbes, including *S. aureus* (MIC, MBC = 63 mg/mL), *E. faecalis* (MIC = 63 mg/mL, MBC = 126 mg/mL), *E. coli* and *P. aeruginosa* (MIC = 31.5 mg/mL, MBC = 126 mg/mL), [68]. Our observations corroborate their findings with the exception of *E. coli* bacteria. However, in the case of *S. cerevisiae*, 100% GTE-specific MBC data are a further novelty.

### 3.4. The Almond GTE Is Effective in Inhibiting E. faecalis, P. vulgaris and L. monocytogenes

The almond GTE has been shown to feature an impressive phytonutrient profile containing a high number of polyphenols (flavonoids and non-flavonoids), amino acids, carboxylic acids and fatty acids. Among the non-flavonoids, there are hydroxycinnamic acids, like chlorogenic acids, caffeic acids, coumaric acids and ferulic acids. The quantitative polyphenol profile showed that the almond GTE contained a high amount of rutoside, hyperoside and chlorogenic acid, and it is likely that these constituents confer some antimicrobial properties [51]. 

Performing the agar diffusion method, it turned out that the almond GTE proved its effectiveness against *E. faecalis* (10.11 ± 0.51 mm inhibition zone) at a 30% concentration and *P. vulgaris* (10.91 ± 0.76 mm inhibition zone) at 40% (see Table 2, Table 3 and Table 7). The almond GTE showed antimicrobial activity for *S. cerevisiae* between 60 and 100% of the GTE concentration. There is no significant difference (*p* = 0.084) between the tested concentrations and the resulting inhibition zones for *B. cereus*, meaning that the 70–80–90–100% concentration solutions could have a greatly similar inhibitory effect. Another study conducted by Ibibia (2013) showed that different concentrations of the extract from Nigerian almond leaves proved effective against *E. coli*, *S. aureus*, *B. subtilis*, *P. aeruginosa* and *B. cereus* [25]. In a different study on the fresh Indian almond leaf alcoholic extract, it showed inhibition for *E. coli* (11 ± 1 mm) and *S. aureus* (17 ± 1) [69]. The antimicrobial properties in terms of microbial specificity are evident when comparing the almond GTE and the leaf extract.

For this extract, *B. cereus* and *P. vulgaris* proved to be the most sensitive in terms of MIC; even a 10% extract concentration exerted a bacteriostatic effect, while the least sensitive was *S. cerevisiae*. The bacterial growth inhibition was detected in the case of *E. faecalis*, *L. monocytogenes*, *P. aeruginosa*, *E. coli* and *S. enterica*. 

Musarra-Pizzo and colab. (2019) investigated the antimicrobial effect of almond peel extract on the hospital strains of *S. aureus*, and in their research, the MIC values ranged from 0.31 to 1.25 mg/mL, while the MBC values were greater than 1.25 mg/mL [70]. Another study examined Italian almond peel; in this case, the MICs were 125 µg/mL for *L. monocytogenes*, 15.62–32.25 µg/mL for *S. aureus*, 500 µg/mL for *E. coli* and 250–500 µg/mL for *P. aeruginosa* while the MBC was greater than 1 mg/mL in all cases [71]. In the case of almond GTE, the most effective bactericidal outcome was seen for *L. monocytogenes* and *E. faecalis.* Our data suggest that the almond GTE and peel extracts have relatively different antimicrobial properties. 

### 3.5. Walnut GTE Shows Broader Antimicrobial Specificity, Excelling against L. monocytogenes and B. cereus

Prior to our research, the antimicrobial properties of various parts of the walnut (*Juglans regia* L.) were investigated by others. Farooqui and colab. (2015) tested the Pakistani walnut bark extract against various antibiotic-resistant microbes, which proved to be effective for the following microbes: methicillin-resistant *S. aureus*, *B. subtilis*, *E. coli*, multidrug-resistant *Salmonella enterica* serovar Typhi, *P. aeruginosa*, *Streptococcus pyogenes*, *Streptococcus pneumoniae*, *Enterobacter cloacae*, *Pasteurella multocida*, *Helicobacter pylori*, *Campylobacter jejuni*, *Shigella species* and *Micrococcus* [5]. In another study conducted by Muzzaffer and Paul (2018), the Indian male *Juglans regia* flower extract was effective against *S. aureus*, *B. subtilis*, *E. coli*, *P. vulgaris*, *Candida albicans* and *Candida glabrata* microbes [72]. 

By means of the agar diffusion method, it was revealed that *L. monocytogenes*, *B. cereus*, *E. faecalis* and *P. vulgaris* were sensitive to different concentrations of walnut GTE (see Table 2, Table 3 and Table 7). It proved most effective against *L. monocytogenes*, where the 100% extract produced an inhibition zone of 21.98 ± 1.13 mm (Figure 1a), and even the 20% concentration had an inhibitory effect showing an inhibition diameter of 13.4 ± 0.53 mm. 

The *B. cereus* was the most sensitive in the case of the MIC assay (MIC = 20%) when the walnut GTE was used, while *S. cerevisiae* was the least sensitive (MIC = 70%) (see Table 5). Furthermore, bacteriostatic effects were also observed for *S. aureus* (MIC = 30%), *L. monocytogenes* (MIC = 30%), *P. vulgaris* (MIC = 30%), *P. aeruginosa* and *E. faecalis*, both with a MIC of 40%, *S. enterica* (MIC = 50%) and *E. coli* (MIC = 50%). The bactericidal effect of the walnut GTE was observed only in the case of *L. monocytogenes* (MBC = 50%), *E. faecalis* (MBC = 90%) and *S. enterica* (MBC = 100%) (see Table 6). Interestingly, a study performed by Vieira and colab. (2019) exhibited that the Portuguese walnut leaf extract was effective against *E. coli* (MIC = 20 mg/mL, MBC > 20 mg/mL), *P. aeruginosa* (MIC and MBC > 20 mg/mL), *E. faecalis* and *L. monocytogenes* (MIC = 2.5 mg/mL, MBC > 20 mg/mL) bacteria [73]. The comparison of walnut GTE and leaf extract bactericidal activity showed a great level of similarity for *L. monocytogenes* and *E. faecalis*.

### 3.6. Black Mulberry GTE Predominantly Inhibits the Growth of B. cereus and P. vulgaris, While Exerting Moderate Bactericidal Activity against L. monocytogenes

Based on the research conducted by Aleya and colab. (2023), it has been shown that the black mulberry GTE contains mostly flavonoids and polyphenols, followed by amino acids and carboxylic acids [51]. Among the 29 non-flavonoid polyphenols, there were hydroxycinnamic acids (chlorogenic, caffeic, coumaric, ferulic) and hydroxybenzoic acids and stilbenes (such as resveratrol). All of these may be contributors to antimicrobial effects, and we used the agar diffusion method to investigate this prospect. Interestingly, only *B. cereus* and *P. vulgaris* bacteria were slightly sensitive to the black mulberry GTE (see Table 2, Table 3 and Table 7). The 100% concentrated GTE resulted in an inhibition zone of 9.96 ± 0.55 mm for *B. cereus* and 10.40 ± 0.37 mm for *P. vulgaris*. Based on the data shown in Table 2, there were no significant differences (*p* = 0.320) between the inhibition zones obtained throughout the 70–100% concentration range in the case of *B. cereus* or the inhibition zones specific for *P. vulgaris* at the 90–100% concentration interval. Amazingly, besides *B. cereus*, it was shown that the Brazilian black mulberry leaf extract was effective against many other microbial species like *E. faecalis*, *E. coli*, *Klebsiella pneumoniae*, *Salmonella choleraesuis*, *Serratia marcescens*, *Shigella flexneri* and *Staphylococcus aureus* [74]. In another study, the Turkish black mulberry fruit extract was effective against *E. coli* and *S. aureus*, while the leaf extract was effective against *Enterobacter aerogenes*, *E. coli*, *Proteus miribalis*, *P. aeruginosa* and *S. aureus* [75]. It seems clear that the black mulberry GTE generates much different antimicrobial spectra then fruit or leaf extracts.

Regarding the MIC concentrations, *B. cereus*, followed by *P. vulgaris*, *L. monocytogenes* and *S. aureus*, proved to be the most sensitive, while *E. faecalis*, *E. coli* and *S. enterica* were the least sensitive. The minimal bactericidal effect of different concentrations was demonstrated for *S. aureus* (MBC = 80%GTE), *E. faecalis* (MBC = 70%GTE), *L. monocytogenes* (MBC = 50%GTE), *S. enterica* (MBC = 100%GTE) and *S. cerevisiae* (MBC = 100%GTE). In the literature, very diverse MIC and MBC values were reported for some leaf extracts; for example, in the case of *S. aureus*, the MIC values varied between 0.156 and 12.5 mg/mL [74,75].

### 3.7. The Blackcurrant GTE Is Less Effective as a Microbial Growth Inhibitor and Bactericide

The agar diffusion method has revealed that the studied blackcurrant GTE does not show any significant antimicrobial effect, having only an effect on *L. monocytogenes* and *S. cerevisiae* (see Table 2 and Table 3). Interestingly, Raiciu and colab. (2010), studying a Romanian blackcurrant GTE showed an inhibitory effect against *S. aureus*, *P. aeruginosa*, *E. coli*, *A. niger* and *Candida albicans* microorganisms [18]. These apparently contradicting data could be explained by the different places of cultivation or the different types of extract production. Our blackcurrant GTE was alcoholic, whereas Raiciu and colab. analysed an aqueous GTE. It has also been shown that the geographical location of the growing area can influence the phytonutrient profile of different plant species; for example, berries from Romania were found to be richer in antioxidant compounds than those from Russia [76]. Similarly, the antioxidant capacity of blackcurrant was found to vary depending on the season and cultivar [46]. In blackcurrant buds, the most common phenolic compounds are gallic acid, hydroxycinnamic acid derivatives, several flavonols (glycosidealcoolics of quercetin, myricetin, kaempferol and isorhamnetin) and dihydroquercetin derivatives [20,48]. According to another study, the most common flavonols in blackcurrant buds are rutin, isoquercetin and astragalin [46,65]. It is predictable that all these compounds could be responsible for the antimicrobial effects. Bendokas and colabs. (2018) investigated the antimicrobial effect of different Lithuanian blackcurrant fruits and found that a 1% concentration had an inhibitory effect on *Rhodotorula rubra*, *Lactococcus lactis*, *Micrococcus luteus*, *S. aureus*, *L. monocytogenes*, *B. cereus*, *Salmonella typhimurium* and *E. coli* microbes [77]. Such a finding suggests that the blackcurrant fruit extract has a much broader antimicrobial spectrum than the GTE.

In the case of blackcurrant, the most sensitive bacteria revealed using the broth MIC assay was *P. vulgaris*, where the 20%GTE showed a bacteriostatic effect, while the least sensitive were *E. faecalis* (no inhibition at all) and *S. enterica* (inhibited only by the 100% extract), (see Table 5). The extract had no bactericidal effect on any of the tested microorganisms (see Table 6). In the research conducted by Paunović and colab. (2022) reported that the MIC values of Serbian blackcurrant fruit and the leaf extracts for *S. aureus*, *E. coli* and *P. vulgaris* varied between 55.82 and 199.21 mg/mL [78]. Also, in a study conducted by Trajković et al. (2023), the MIC of blackcurrant-lyophilised juice was 100 mg/mL for *S. aureus*, *E. faecalis*, *B. cereus*, *L. monocytogenes*, *E. coli* and *P. aeruginosa*, while the MBC was greater than 100 mg/mL [79]. These previously published studies strongly support the fact that blackcurrant fruit and GTE have reduced antimicrobial activity.

### 3.8. Comparative Evaluation of GTEs Revealing Variable Antimicrobial Activity Strength Levels

Based on obtained data, it was concluded that many of the studied GTEs showed a different kind of antimicrobial effectiveness (see also Table 7). The agar diffusion method (ADM) used is an excellent tool for the initial stage of the antimicrobial screening of GTEs, although it does not provide insight into the antimicrobial mechanism of action. Based on the ADM, the olive GTE showed 5 inhibitory microbial effects, while the almond, bilberry and blackberry GTEs featured 4, followed by walnut with 3, and the black mulberry and blackcurrant with 2 inhibitory microbial effects regarding the tested microbial strains. It was also interesting to note that when the antimicrobial effects of different GTE mixtures were assessed using ADM, the combination of individual olive, walnut and almond GTEs with others could produce synergistic interactions (see Table 4). This is a novel feature of such extracts and one that merits further analysis.

Bacteriostatic and bactericidal effects are considered to be two distinct types of antimicrobial properties based on different mechanisms. Bacteriostatic antibiotics inhibit the reproduction of microorganisms (they are kept in a stationary growth phase), and the latter is removed by the host’s immune system, while bactericidal antibiotics kill the microorganisms. MICs provide a clearer indication of the microbial susceptibility, including resistance and the bacteriostatic efficacy of an antimicrobial, while the MBC indicates the direct lethal effect of the antimicrobial. Bacteriostatic antimicrobials are expected to inhibit bacterial protein synthesis, but this could affect transmembrane potential or interfere with antimicrobial resistance [80]. The MIC assay revealed the olive GTE to be the most bacteriostatic-effective (6 inhibited strains), followed by the blackberry, walnut and bilberry GTEs (with 4 restricted strains), then almond and blackberry GTEs (with 2 strains). Regarding the microbial spectrum’s wideness, the less effective bacteriostatic result was specific to the blackcurrant GTE (limiting only one strain), while all these effects were generated at the lowest 10–30% GTE concentrations. Walnut and almond GTEs proved less effective in the case of the ADM method but presented a better outcome in the MIC assay. These extracts showed lower bactericidal effectiveness, with 6 microorganisms on which no effect could be measured. The black mulberry and blackcurrant GTEs performed worse in the ADM assay but displayed higher effectiveness in the MIC assay. 

In the MBC assay, bactericidal effects were relevant for the olive, bilberry and blackberry GTEs, but the almond and black mulberry GTEs also induced some lethality, while the blackcurrant extract GTE showed no life-limiting detrimental effect.

The observed results also provide a broader perspective on the antimicrobial applicability of the GTEs analysed. In fact, although some are more effective than others, all GTEs are able to inhibit the growth of the microbial species studied, but their effect is concentration-dependent (see Table 7). 

The olive GTE emerged as an efficient antimicrobial with a remarkable bacteriostatic effect for *B. cereus*, followed by *S. aureus* and *E. faecalis*, while the most relevant bactericidal property was seen in the case of *S. aureus* and *E. faecalis*. The growth inhibition of *S. enterica* was also evident. Our study corroborates the recently published findings of Popović et al. [16] and extends our knowledge with respect to species like *P. vulgaris*, *P. aeruginosa*, *S. enterica* and *S. cerevisiae*. The *S. aureus* is an opportunistic pathogen that can lead to different acute or chronic diseases and difficult-to-treat infections in hospitals and other community locations. The *E. faecalis* has been seen in most healthy individuals but could also cause endocarditis, urinary tract infections, meningitis and eventually sepsis. The *S. enterica* has many serovars that might bring about gastrointestinal diseases. 

The walnut GTE showed a significant sensitivity and prominent bacteriostatic effect towards *B. cereus* and *L. monocytogenes*, while the bactericidal property was almost absent (Table 7). This suggests the possible use of the walnut GTE as an enhancer for classical antibiotic treatments to increase efficacy. 

The bilberry GTE featured significant bacteriostatic properties, but a substantial bactericidal effect was displayed mostly for *E. faecalis* and *L. monocytogenes.* The *L. monocytogenes* is considered a virulent foodborne pathogen-inducing neuro- or pregnancy-related listeriosis leading to meningitis [81]. It is also important to mention that the bilberry GTE displayed some specificity towards *P. vulgaris*, which is an opportunistic pathogen of humans causing resistant hospital infections. 

The blackberry GTE proved itself as a potent antimicrobial with a broad bacteriostatic but a more specific bactericidal effect on *L. monocytogenes* and *S. aureus*. Our observations suggest the putative suitability of bilberry and blackberry GTEs for the treatment of listeriosis, although the bilberry–blackberry GTE mixture did not show any interacting property. 

The almond and black mulberry GTEs emerged as more bacteriostatic than bactericide (see Table 6), displaying some kind of specificity towards *L. monocytogenes*, and noticeably, their mixture does not reveal any interacting features. 

Among the most interesting observations reported in this study are the mixtures of some GTEs that showed synergistic antimicrobial interactions. Mixtures such as blueberry–almond, walnut–black mulberry and walnut–blackcurrant are combinations of GTEs that showed effective antimicrobial activity against *S. aureus*. It is noteworthy that the individual GTEs mentioned above did not inhibit the growth of *S. aureus*, while the mixture of two GTEs resulted in the formation of some kinds of molecular hybrids that could gain antimicrobial properties and might be used for the development of novel hybrid antibiotics [82,83].

Taken together, comparing the ADM with the MIC and MBC broth assays, it could be seen that the studied microorganisms were much more sensitive to being suspended in broth. The analysis of MIC and MBC facilitated the comparison of bacteriostatic and bactericidal effects. It has been suggested that bactericidal activity at lower concentrations, as with olive, bilberry, or blackberry GTEs, is beneficial for a new antibacterial factor, inhibiting the emergence of resistance by preventing bacterial regrowth [84]. Furthermore, in clinical practice, the categorisation of antibiotics into bacteriostatic and bactericidal is debatable in situations concerning abdominal, skin and soft tissue infections and pneumonia [85]. In the case of pneumonia, it was demonstrated that bactericidal effects are not statistically different compared with bacteriostatic antibiotics in clinical cure rates, treatment failure, or relapse rates [86]. This could imply that an important element of personalised antimicrobial treatment should be the application of a decision regarding the possible superiority of bacteriostatic over bactericidal antibiotics or vice versa. In case the bacteriostatic effect is more favoured over the bactericidal property, the olive, bilberry, black mulberry, walnut and blackcurrant GTEs should be further studied in the context of enteritis. The olive, black mulberry, blackcurrant and blueberry GTEs have been shown to contain many phytonutrients with expected or proven anti-inflammatory properties, so they can have a synergistic healing effect by limiting microbial growth and overcoming inflammation. The quest for new antimicrobials continues, and as the scientific considerations become more diverse, looking for natural solutions seems to be an endless source of inspiration.

## 4. Materials and Methods

### 4.1. The Gemmotherapy Extracts

The reported research is based on seven different GTEs that correspond to species like *Olea europaea*, *Prunus amygdalus*, *Morus nigra*, *Juglans regia*, *Rubus fruticosus*, *Ribes nigrum* and *Vaccinium myrtillus.* The buds and young shoots were collected from different places at different times. *Olea europaea* young shoots and *Prunus amygdalus* buds were harvested from a plantation in Calabria, Italy, in June and April 2022, respectively; *Morus nigra*, *Juglans regia* and *Ribes nigrum* buds were obtained from the organic plantations of the PlantExtrakt company, collected in March–April 2022; *Rubus fruticosus* and *Vaccinium myrtillus* young shoots were collected from wild flora in the Mărișel area, Cluj county, Romania, in June 2022. The fresh vegetal material was processed at a maximum of 6 h from collection or stored in the refrigerator at 4 °C for a maximum of 24 h until manufacturing. 

### 4.2. Sample Preparation and Extraction

The extraction method is based on the method written in European Pharmacopoeia, monograph 07/2022:2371, method 2.1.3. The extracts were prepared from freshly harvested vegetal materials that were preserved in a 1:1 mixture of 96% (*v*/*v*) ethanol and glycerol, with a plant-to-solvent ratio of 1:2. A moisture content analysis was performed on the fresh samples. Based on the determined moisture content, the solvent quantity was calculated to achieve a ratio of dry plant-to-solvent at 1:20. The solvent was a 1:1 mixture of 96% (*v*/*v*) ethanol and glycerol. The obtained plant–solvent mixture was mixed periodically for 20 days, 2 × 20 min/day. For the next step, the solid and the liquid parts of the mixture were separated, and the extracted solid plant material was further pressed to increase the yield of extraction. The two extracted solutions (separated from the solid part and those obtained after the pressing of the solid part) were mixed, forming the concentrated extracts that were used in further studies. 

### 4.3. Studied Microorganisms 

The reference bacterial and microscopic fungi strains used in this study were obtained from the National Collection of Agricultural and Industrial Microorganisms (NCAIM). The determination of antimicrobial activity of different GTEs was carried out on the following eight bacteria strains: *Escherichia coli* B.00200, *Pseudomonas aeruginosa* B.01064, *Salmonella enterica* subsp. *enterica* B.00834, *Proteus vulgaris* B.00642 (Gram-negative bacteria); *Bacillus cereus* B.00076, *Staphylococcus aureus* B.01055 and *Enterococcus faecalis* B.01054 (Gram-positive bacteria). It was also carried out on five mycotoxigenic fungi, including *Aspergillus flavus* F.00048, *A. niger* F.00071, *A. ochraceus* F.00850, *Penicillium citrinum* F.00815, *P. expansum* F.00601 and one yeast strain *Saccharomyces cerevisiae* Y.00481. Bacterial strains were cultivated on nutrient agar (peptone 10 g, meat extract 10 g, NaCl 5 g, agar 18 g, distilled water 1000 mL) at 37 °C for 24 h, moulds and yeast were cultivated on a complex medium (peptone 10 g, yeast extract 10 g, glucose 40 g, agar 20 g, distilled water 1000 mL) at 28 °C for 72 h, obtained from VWR International L.L.C. (Debrecen, Hungary). 

### 4.4. Antimicrobial Activity Assessment

The GTE-specific antimicrobial effects were studied using the agar-well diffusion method. Prior to analysis, ethanol was removed from the extracts using a rotavapor, operated at a maximum of 40 °C degrees and at 200 mbar, avoiding in these conditions the degradation and loss of the bioactive compounds. The removed ethanol was replaced immediately after evaporation with purified water. The obtained samples were stored in the refrigerator at 4 °C degrees until the study was performed. 

A set of GTE concentrations from 0 to 100% (*v*/*v*) was obtained, where 100% corresponded to the concentrated GTE, and the rest were diluted with sterile distilled water. The concentrations of the diluted GTE/set were as follows: 100%–50 mg/mL, 90%–45 mg/mL, 80%–40 mg/mL, 70%–35 mg/mL, 60%–30 mg/mL, 50%–25 mg/mL, 40%–20 mg/mL, 30%–15 mg/mL, 20%–10 mg/mL, 10%–5 mg/mL. Based on their effects on a given microorganism, a few of the concentrated solutions were also mixed together in a 1:1 ratio and tested as a GTE mixture combination for effectiveness.

The bacterial and fungal suspension of 1 OD (optical density) was prepared in a turbidity tube, from which 0.1 mL of microbial suspension (10^8^ CFU/mL) was inoculated on the surface of the nutrient medium. After this, an 8 mm diameter hole was cut in the centre of the medium, into which 0.1 mL of the extract was pipetted into different concentrations. After incubation for 24 h at 37 °C for *Bacillus cereus*, *Escherichia coli*, *Pseudomonas aeruginosa*, *Salmonella enterica* subsp. *enterica*, *Proteus vulgaris*, *Staphylococcus aureus*, *Enterococcus faecalis* and *Listeria monocytogenes*, the diameter of the inhibition zones (together with the hole) was measured using a digital calliper [87]. To accurately evaluate the obtained results, the average of three parallel measurements was calculated. The same method was used for yeast and moulds (*Saccharomyces cerevisiae*, *Aspergillus niger*, *Aspergillus flavus*, *Aspergillus ochraceus*, *Penicillium citrinum*, *Penicillium expansum*), except that the medium was a complex medium and the incubation was performed at 28 °C for 48 h.

### 4.5. Broth Microdilution Method

The antimicrobial assay was performed using the broth microdilution method with 96 well plates. Each of the stock-concentrated GTE samples was diluted serially in the microplate wells to obtain 100 μL of the mixed solution using nutrient broth as the medium for dilution. A concentration range between 10 and 100% (concentrations in mg/mL are mentioned above) was achieved as a result. 

### 4.6. Minimum Inhibitory Concentration Assay (MIC)

The MICs were determined according to a method described by Agbeby et al. (2022) [88] and El Baabouaa et al. (2022) [49]. The overnight culture of the tested microorganism was diluted to 1 OD, which is equivalent to an inoculum size of 1.0 × 10^8^ CFU/mL. Each well of the microtiter plate contained 100 μL of various concentrations of the GTEs (as explained above), and 20 μL of the tested microorganisms (eight bacteria and one yeast) were inoculated into them. The bacteria-containing microplates were incubated at 37 °C, while the yeast-containing microplates were incubated at 28 °C for 24 h. After incubation, 10 μL of 0.01 mg/mL resazurin was added to each well and incubated for another 2 h and microbial growth was revealed by a change in colouration from purple to pink. The lowest concentration of each extract with no visible growth was recorded as the minimum inhibitory concentration (MIC) against the respective microbial isolates. As a verifying method, samples were inoculated on a nutrient medium from several wells, and the colonies formed were counted to determine the authenticity of the colour change.

### 4.7. Minimum Bactericidal Concentration Assay (MBC)

The MBC was determined using the microtiter plate method used in the MIC determination. The nutrient broth in the wells of the microtiter plate, which did not show any growth after incubation during the MIC assays was diluted and spread separately on nutrient agar and incubated in an inverted position at 37 °C for 24 h (bacteria) and at 28 °C for 48 h (yeast). The MBC was regarded as the lowest concentration of the extract, which did not produce any growth on the nutrient agar after 24–48 h of incubation.

### 4.8. The Phytochemical Analysis

The LC/MS method was performed on a Shimadzu Nexera I LC/MS—8045 (Kyoto, Japan) UHPLC system equipped with a quaternary pump and autosampler, an ESI probe and quadrupole rod mass spectrometer, respectively [51].

The separation was carried out on a Luna C18-reversed phase column (150 mm × 4.6 mm × 3 mm, 100 Å) from Phenomenex (Torrance, CA, USA). The column was maintained at 40 °C degrees during the analyses. 

The mobile phase (see Figure 5) was a gradient made from methanol (Merck, Darmstadt, Germany) and ultra-purified water prepared using the Simplicity Ultra-Pure Water Purification System (Merck Millipore, Billerica, MA, USA). As an organic modifier, formic acid (Merck, Darmstadt, Germany), in the form of a 2% solution in ultra-purified water, methanol and formic acid was of the LC/MS grade. The used flow rate was of 0.5 mL/minute. The total time of the analysis was 35 min.

The detection was performed on a quadrupole rod mass spectrometer operated with electrospray ionisation (ESI), both in the negative and positive MRM (multiple reaction monitoring) ion mode (see Table 8). The interface temperature was set at 300 °C degrees. For vaporisation, drying gas nitrogen was used at 35 psi and 10 mL/min, respectively. The capillary potential was set at +3000 V.

The substances from Table 9 were used as the standards. From each standard at each concentration, 1 µL was injected. The identification was performed by a comparison of MS spectra and their transitions between the separated compounds and standards (see Table 8). The identification and quantification were made basely on the main transition from the MS spectra of the substance. For the purpose of quantification, the calibration curves were determined (the equations for which are given in Table 9).

### 4.9. Statistical Analysis

All analyses were performed in triplicate, and data were then expressed as the mean ± standard deviation (SD). The statistical analysis of the data was performed using IBM SPSS Statistics 26. For antimicrobial activity, the data were subjected to the statistical analysis of variance (ANOVA-1) followed by Tukey’s HSD test to evaluate the significant differences between various concentrations and extracts. The difference was regarded as significant when *p* < 0.05.

## 5. Conclusions

The current study aims to elucidate the antimicrobial properties of seven GTEs, and for five of them, the reported phytonutrient profiles suggest putative antidiabetic, anti-inflammatory and antimicrobial implications. The comparative nature of this study permits us to directly compare GTEs and even draw conclusions about their relative antimicrobial strength. Interestingly, the agar diffusion method revealed the blackberry (*R. fruticosus*) GTE to inhibit the growth of eight microbial species among the tested ones and, quite remarkably, included the mould and yeast strains. The blackberry GTE emerges as a potent antimicrobial that inhibits the growth of many microorganisms, while the microbial death-inducing effect is more relevant for *S. aureus* and *L. monocytogenes*. Furthermore, the olive (*O. europaea* L.) GTE appeared to feature the strongest bacteriostatic and bactericidal effects with increased specificity for *S. aureus*, *E. faecalis* and *L. monocytogenes*. Next to olive, the bilberry (*V. myrtillus*) GTE showed increased bacteriostatic but diminished bactericidal properties with some specificity towards *E. faecalis* and *L. monocytogenes*. Walnut (*J. regia*) and black mulberry (*M. nigra*) GTEs featured similar bacteriostatic and bactericidal strengths, while the walnut GTE looked more specifically towards *E. faecalis* and *L. monocytogenes* which were shortly followed by *B. cereus* and *P. vulgaris*. The black mulberry (*M. nigra*) GTE presented further bacteriostatic specificity towards *L. monocytogenes* and then *B. cereus* and *P. vulgaris*, on which the GTE did not induce any bactericide effect. The almond (*P. amygdalus*) GTE brought about a weaker bacterial growth inhibition that was completed by a slight death induction for *E. faecalis*. Finally, the blackcurrant (*R. nigrum*) GTE had very little effect on bacterial growth or death, indicating that this GTE had no significant antimicrobial activity. Taken together, with the exception of blackcurrant, all the other GTEs have a more significant inhibition of bacterial growth than the induction of microbial death, which is a feature that suggests their applicability to enhance the antimicrobial effects of other antibiotics under less stringent conditions.

## Figures and Tables

**Figure 1 antibiotics-13-00181-f001:**
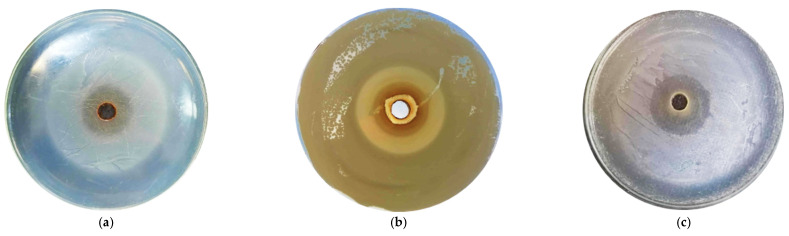
The different sizes of the inhibition zones induced by the GTEs on some microorganisms: (**a**) *L. monocytogenes*—*Juglans regia* 100% GTE; (**b**) *P. vulgaris*—*Vaccinium myrtillus* 100% GTE; and (**c**) *L. monocytogenes*—*Rubus fruticosus* 100% GTE.

**Figure 2 antibiotics-13-00181-f002:**
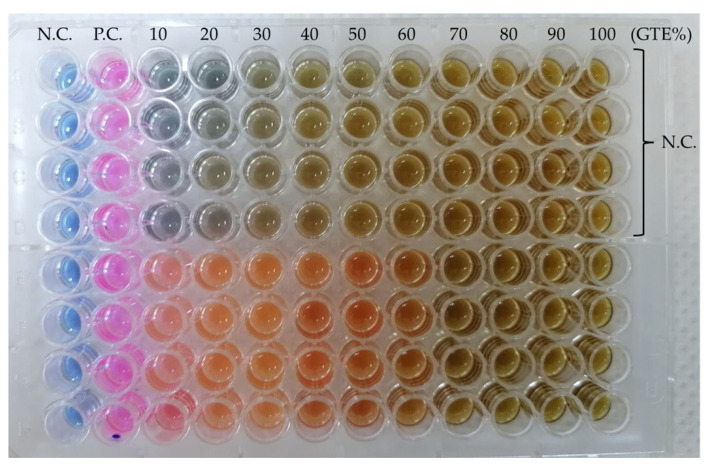
Picture of a microtiter plate with the various concentrations of GTEs and the microorganism in broth culture. Pink to red colour indicates bacterial growth. (N.C.—negative control, P.C.—positive control).

**Figure 3 antibiotics-13-00181-f003:**
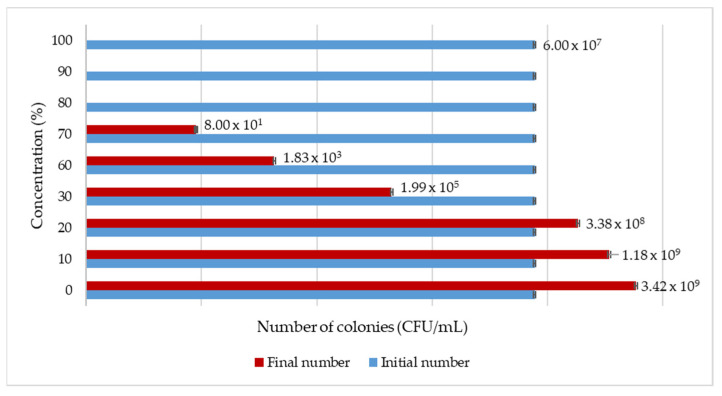
Olive GTE concentration with specific *Salmonella enterica* colony numbers.

**Figure 4 antibiotics-13-00181-f004:**
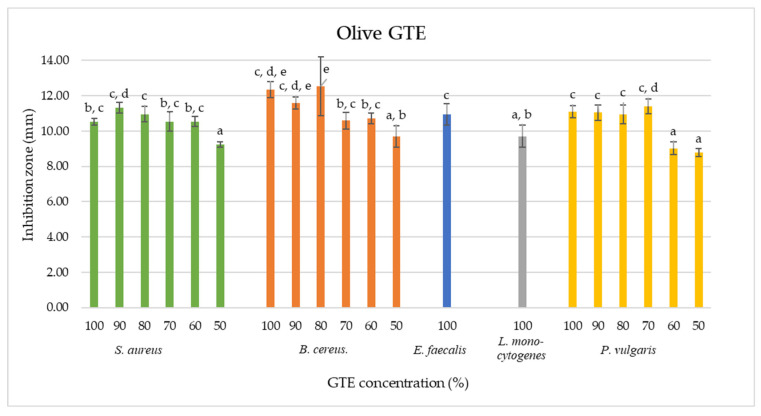
Inhibition zone size comparison for *Olea europaea.* Values with different letters (^a–e^) are statistically different at *p* < 0.05, according to Tukey’s test.

**Figure 5 antibiotics-13-00181-f005:**
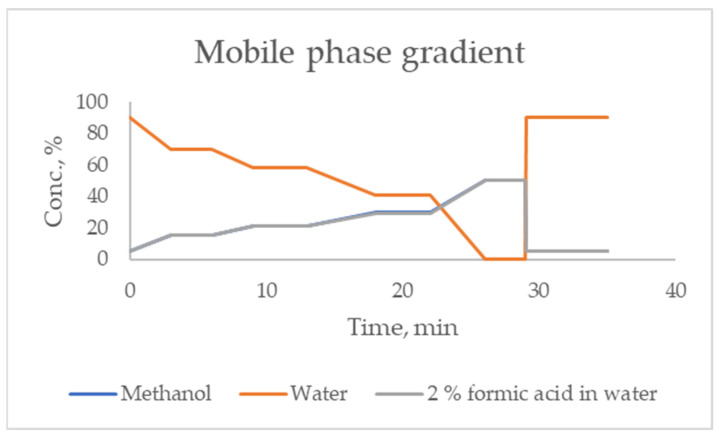
The mobile phase gradient.

**Table 1 antibiotics-13-00181-t001:** The phytochemical composition of GTEs.

Studied Components	OGTE	AGTE	BMGTE	WGTE	BBGTE	BkBGTE	BCGTE
Phenolic acids							
*Caffeic acid*	-	0.825 ± 0.0094	-	-	1.693 ± 0.0188	-	1.693 ± 0.0101
*Chlorogenic acid*	0.265 ± 0.0042	1.390 ± 0.0095	3.539 ± 0.0251	0.244 ± 0.0038	7.552 ± 0.0217	0.157 ± 0.0057	0.227 ± 0.0057
*Ferulic acid*	-	-	-	-	-	-	0.109 ± 0.0086
*Gallic acid*	-	-	-	-	-	0.049 ± 0.0010	0.049 ± 0.0008
*Salicylic acid*	0.053 ± 0.0012	-	-	-	0.066 ± 0.0009	0.895 ± 0.0202	0.071 ± 0.0017
Flavonoids							
*Apigenin*	0.055 ± 0.0021	0.017 ± 0.0009	0.103 ± 0.0025	0.002 ± 0.0001	-	0.330 ± 0.0108	0.043 ± 0.0011
*Catechin*	-	-	-	0.008 ± 0.0001	0.044 ± 0.0018	-	0.028 ± 0.0009
*Chrysine*	0.109 ± 0.0051	0.103 ± 0.0049	0.093 ± 0.0009	-	0.117 ± 0.0085	0.101 ± 0.0022	0.114 ± 0.0027
*Hyperoside*	0.202 ± 0.0074	1.967 ± 0.0157	0.162 ± 0.0052	1.301 ± 0.0094	0.392 ± 0.0102	0.172 ± 0.0089	0.547 ± 0.0187
*Kaempferol*	-	0.032 ± 0.0010	-	-	0.033 ± 0.0009	-	-
*Luteolin*	0.049 ± 0.0009	-	0.017 ± 0.0014	-	-	0.013 ± 0.0008	-
*Luteolin-7-O-glucoside*	1.777 ± 0.0257	-	0.072 ± 0.0023	0.072 ± 0.0023	-	0.078 ± 0.0012	0.074 ± 0.0021
*Naringenin*	0.032 ± 0.0011	0.110 ± 0.0067	0.046 ± 0.0014	0.064 ± 0.0011	0.036 ± 0.0005	0.043 ± 0.0009	-
*Quercetin*	0.052 ± 0.0015	0.201 ± 0.0076	-	0.313 ± 0.0092	0.989 ± 0.0118	-	0.210 ± 0.0100
*Rutoside*	0.416 ± 0.0201	5.506 ± 0.0051	1.367 ± 0.0204	0.103 ± 0.0024	0.105 ± 0.0028	0.278 ± 0.0047	1.662 ± 0.0198

The concentrations are expressed in mg/mL, mean ± RSD.

**Table 2 antibiotics-13-00181-t002:** The antimicrobial effects of GTEs, concentrations and inhibition zones (*n* = 3).

Studied Microorganisms	Conc. (%)	OGTE	AGTE	BMGTE	WGTE	BkBGTE	BBGTE	BCGTE
Gram-positive bacteria								
*B. cereus*	100	12.34 ± 0.46 ^d,e^	10.17 ± 0.92 ^a,b,c,d^	9.96 ± 0.55 ^a^	13.19 ± 1.17 ^d,e^	nd	10.45 ± 0.55 ^c,d^	nd
	90	11.59 ± 0.34 ^c,d,e^	9.71 ± 0.30 ^a,b^	9.67 ± 0.69 ^a^	11.79 ± 0.83 ^b,c,d^	nd	10.70 ± 0.85 ^c,d,e,f^	nd
	80	12.54 ± 1.65 ^e^	9.40 ± 0.28 ^a^	10.18 ± 0.73 ^a^	11.74 ± 0.59 ^b,c,d^	nd	10.56 ± 0.65 ^c,d,e^	nd
	70	10.59 ± 0.46 ^b,c^	9.53 ± 0.27 ^a,b^	9.70 ± 0.54 ^a^	10.95 ± 0.82 ^a,b^	nd	9.91 ± 0.57 ^a,b,c^	nd
	60	10.73 ± 0.30 ^b,c^	nd	nd	11.29 ± 0.49 ^a,b,c^	nd	nd	nd
	50	9.68 ± 0.60 ^a,b^	nd	nd	10.22 ± 0.52 ^a,b^	nd	nd	nd
	40	nd	nd	nd	10.18 ± 0.33 ^a^	nd	nd	nd
	30	nd	nd	nd	nd	nd	nd	nd
*S. aureus*	100	10.52 ± 0.18 ^b,c^	nd	nd	nd	12.2 ± 0.51 ^c,d^	nd	nd
	90	11.32 ± 0.31 ^c,d^	nd	nd	nd	10.47 ± 0.40 ^a^	nd	nd
	80	10.96 ± 0.43 ^c^	nd	nd	nd	10.79 ± 1.06 ^a,b,c^	nd	nd
	70	10.54 ± 0.55 ^b,c^	nd	nd	nd	13.95 ± 0.63 ^e,f^	nd	nd
	60	10.55 ± 0.27 ^b,c^	nd	nd	nd	13.29 ± 0.65 ^d,e^	nd	nd
	50	9.24 ± 0.15 ^a^	nd	nd	nd	13.22 ± 0.58 ^d,e^	nd	nd
	40	nd	nd	nd	nd	9.81 ± 0.56 ^a^	nd	nd
	30	nd	nd	nd	nd	nd	nd	nd
*E. faecalis*	100	10.95 ± 0.60 ^c^	13.86 ± 0.46 ^m,n^	nd	10.99 ± 0.91 ^a,b,c^	nd	10.91 ± 0.46 ^d,e,f^	nd
	90	nd	14.17 ± 0.55 ^n^	nd	10.53 ± 0.97 ^a,b^	nd	10.49 ± 0.63 ^c,d^	nd
	80	nd	13.30 ± 0.29 ^m,n^	nd	10.09 ± 0.59 ^a^	nd	9.97 ± 0.18 ^a,b,c,d^	nd
	70	nd	13.03 ± 0.42 ^k,l,m^	nd	nd	nd	10.00 ± 0.20 ^a,b,c,d^	nd
	60	nd	12.88 ± 0.28 ^k,l^	nd	nd	nd	9.84 ± 0.26 ^a,b,c^	nd
	50	nd	12.75 ± 0.55 ^j,k,l^	nd	nd	nd	10.14 ± 0.42 ^b,c,d^	nd
	40	nd	11.88 ± 0.56 ^h,i,j^	nd	nd	nd	nd	nd
	30	nd	10.11 ± 0.51 ^a,b,c,d^	nd	nd	nd	nd	nd
	20	nd	nd	nd	nd	nd	nd	nd
*L. monocytogenes*	100	9.71 ± 0.62 ^a,b^	nd	nd	21.98 ± 1.13 ^i^	19.20 ± 0.87 ^i^	nd	10.81 ± 0.74 ^b^
	90	nd	nd	nd	17.16 ± 0.90 ^g,h^	18.71 ± 0.60 ^h,i^	nd	10.77 ± 0.41 ^b^
	80	nd	nd	nd	18.08 ± 0.41 ^h^	18.59 ± 0.41 ^h,i^	nd	nd
	70	nd	nd	nd	15.53 ± 0.83 ^f,g^	17.84 ± 0.71 ^g,h,i^	nd	nd
	60	nd	nd	nd	16.73 ± 0.69 ^g,h^	17.37 ± 2.31 ^g,h^	nd	nd
	50	nd	nd	nd	14.13 ± 0.47 ^e,f^	17.33 ± 0.46 ^g,h^	nd	nd
	40	nd	nd	nd	15.07 ± 1.46 ^f^	16.93 ± 0.68 ^g^	nd	nd
	30	nd	nd	nd	13.33 ± 0.49 ^d,e^	15.4 ± 0.73 ^f^	nd	nd
	20	nd	nd	nd	12.61 ± 0.96 ^c,d,e^	13.4 ± 0.53 ^d,e^	nd	nd
	10	nd	nd	nd	nd	nd	nd	nd
Gram-negative bacteria								
*P. vulgaris*	100	11.09 ± 0.33 ^c^	11.14 ± 0.99 ^e,f,g,h^	10.40 ± 0.37 ^a,b^	10.33 ± 0.34 ^a,b^	12.77 ± 0.64 ^d,e^	12.76 ± 0.80 ^h,i^	nd
	90	11.05 ± 0.44 ^c^	12.12 ± 0.84 ^i,j,k^	10.53 ± 0.58 ^a,b^	nd	11.18 ± 0.45 ^a,b,c^	13.55 ± 0.75 ^i,j^	nd
	80	10.94 ± 0.52 ^c^	11.57 ± 0.73 ^f,g,h,i^	11.52 ± 1.31 ^b^	nd	10.99 ± 0.30 ^a,b,c^	15.04 ± 1.03 ^k^	nd
	70	11.40 ± 0.42 ^c,d^	10.69 ± 0.64 ^c,d,e,f^	nd	nd	nd	14.24 ± 0.86 ^j,k^	nd
	60	9.02 ± 0.37 ^a^	10.00 ± 0.71 ^a,b,c,d^	nd	nd	nd	11.91 ± 0.48 ^g,h^	nd
	50	8.78 ± 0.22 ^a^	10.44 ± 0.33 ^b,c,d,e^	nd	nd	nd	10.59 ± 0.38 ^c,d,e^	nd
	40	nd	10.91 ± 0.76 ^d,e,f,g^	nd	nd	nd	11.64 ± 0.57 ^f,g^	nd
	30	nd	nd	nd	nd	nd	10.15 ± 0.45 ^b,c,d^	nd
	20	nd	nd	nd	nd	nd	nd	nd
*P. aeruginosa*	100	nd	nd	nd	nd	nd	nd	nd
*E. coli*	100	nd	nd	nd	nd	nd	nd	nd
*S. enterica*	100	nd	nd	nd	nd	nd	nd	nd
Yeast								
*S. cerevisiae*	100	nd	11.76 ± 0.66 ^g,h,i^	nd	nd	10.39 ± 0.43 ^a^	10.84 ± 0.37 ^c,d,e,f^	9.63 ± 0.35 ^a^
	90	nd	11.84 ± 0.14 ^g,h,i,j^	nd	nd	11.97 ± 0.64 ^b,c,d^	11.68 ± 0.9 ^f,g^	10.61 ± 0.96 ^a,b^
	80	nd	10.20 ± 0.16 ^a,b,c,d,e^	nd	nd	10.43 ± 0.42 ^a^	11.52 ± 0.56 ^e,f,g^	10.90 ± 0.20 ^b^
	70	nd	9.95 ± 0.28 ^a,b,c^	nd	nd	10.56 ± 0.55 ^a,b^	9.90 ± 0.48 ^a,b,c^	9.78 ± 0.38 ^a^
	60	nd	9.43 ± 0.13 ^a^	nd	nd	nd	nd	nd
	50	nd	nd	nd	nd	nd	nd	nd
Moulds								
*A. niger*	100	nd	nd	nd	nd	nd	nd	nd
*A. flavus*	100	nd	nd	nd	nd	10.4 ± 0.27	nd	nd
	90	nd	nd	nd	nd	10.13 ± 0.25	nd	nd
	80	nd	nd	nd	nd	9.76 ± 0.54	nd	nd
	70	nd	nd	nd	nd	9.42 ± 0.25	nd	nd
	60	nd	nd	nd	nd	nd	nd	nd
*A. ochraceus*	100	nd	nd	nd	nd	10.47 ± 0.7	nd	nd
	90	nd	nd	nd	nd	10.07 ± 0.26	nd	nd
	80	nd	nd	nd	nd	9.96 ± 0.22	nd	nd
	70	nd	nd	nd	nd	9.56 ± 0.19	nd	nd
	60	nd	nd	nd	nd	nd	nd	nd
*P. citrinum*	100	nd	nd	nd	nd	14.02 ± 0.64	9.34 ± 0.25	nd
	90	nd	nd	nd	nd	13.22 ± 0.32	9.09 ± 0.31	nd
	80	nd	nd	nd	nd	12.81 ± 0.36	8.83 ± 0.35	nd
	70	nd	nd	nd	nd	11.91 ± 0.24	9.00 ± 0.18	nd
	60	nd	nd	nd	nd	nd	nd	nd
*P. expansum*	100	nd	nd	nd	nd	9.09 ± 0.07	nd	nd
	90	nd	nd	nd	nd	8.87 ± 0.19	nd	nd
	80	nd	nd	nd	nd	8.90 ± 0.22	nd	nd
	70	nd	nd	nd	nd	8.83 ± 0.14	nd	nd
	60	nd	nd	nd	nd	nd	nd	nd

nd—not detectable. Results are expressed as the mean in mm ± SD. Inhibition zones, including the diameter of the hole (8 mm). Values with different letters (^a–n^) within a column are statistically different at *p* < 0.05, according to Tukey’s test.

**Table 3 antibiotics-13-00181-t003:** The GTEs specific minimal antimicrobial concentration (%) revealed by the agar diffusion method.

Studied Microorganisms	OGTE	AGTE	BMGTE	WGTE	BBGTE	BkBGTE	BCGTE
Gram-positive bacteria							
*B. cereus*	50	70	70	40	70	nd	nd
*S. aureus*	50	nd	nd	nd	nd	40	nd
*E. faecalis*	100	30	nd	80	50	nd	nd
*L. monocytogenes*	100	nd	nd	20	nd	20	90
Gram-negative bacteria							
*P. vulgaris*	50	40	80	100	30	80	nd
*P. aeruginosa*	nd	nd	nd	nd	nd	nd	nd
*E. coli*	nd	nd	nd	nd	nd	nd	nd
*S. enterica*	nd	nd	nd	nd	nd	nd	nd
Yeast							
*S. cerevisiae*	nd	60	nd	nd	70	70	70
Moulds							
*A. niger*	nd	nd	nd	nd	nd	nd	nd
*A. flavus*	nd	nd	nd	nd	70	70	nd
*A. ochraceus*	nd	nd	nd	nd	nd	70	nd
*P. citrinum*	nd	nd	nd	nd	nd	70	nd
*P. expansum*	nd	nd	nd	nd	nd	70	nd

nd—non-detectable.

**Table 4 antibiotics-13-00181-t004:** Mixed GTEs tested antimicrobial effects on selected bacteria.

Extract Mixture	*B. cereus*	*S. aureus*	*E. faecalis*	*L. monocytogenes*	*P. vulgaris*	*P. aeruginosa*	*E. coli*	*S. enterica*
OGTE + WGTE	10.07 ± 0.61 +	nt	nt	nt	nt	nt	nt	nt
BCGTE + AGTE	9.16 ± 0.22 ++	nt	nt	nt	nt	nt	nt	-
BBGTE + WGTE	12.1 ± 0.66 +	nt	nt	nt	11.99 ± 0.3 +	-	-	nt
BBGTE + AGTE	9.48 ± 0.29 ++	16.57 ± 0.27++	-	nt	12.54 ± 0.28 +	nt	nt	nt
BBGTE + BMGTE	10.66 ± 0.33 ++	nt	nt	nt	nt	nt	nt	nt
OGTE + AGTE	11.13 ± 0.36 +	16.91 ± 1.02 ++	nt	nt	nt	-	-	-
BMGTE + AGTE	8.67 ± 0.1 ++	nt	nt	-	nt	-	-	nt
OGTE + BCGTE	-	nt	-	-	nt	nt	-	-
WGTE + BMGTE	-	13.17 ± 0.8 ++	nt	15.2 ± 0.28 +	nt	-	-	-
OGTE + BMGTE	-	nt	nt	nt	nt	nt	nt	nt
BCGTE + BkBGTE	-	nt	nt	nt	nt	-	-	-
OGTE + BkBGTE	-	12.78 ± 0.26 +	nt	nt	nt	-	-	-
WGTE + AGTE	-	nt	-	nt	nt	nt	nt	nt
WGTE + BCGTE	-	10.87 ± 0.58 ++	-	nt	-	nt	nt	nt
BBGTE + OGTE	nt	12.17 ± 0.43 +		-	nt	-	nt	nt
BkBGTE + BMGTE	nt	12.74 ± 0.96 +	-	14.86 ± 0.4 +	nt	-	nt	nt
WGTE + BkBGTE	nt	14.45 ± 0.35 +		16.2 ± 0.36 +	nt	nt	-	-
BBGTE + BCGTE	nt	-		-	11.52 ± 0.45 +	nt	nt	nt
BCGTE + BMGTE	nt	nt	-	nt	nt	-	-	-
WGTE + OGTE	nt	nt	-	10.7 ± 0.14 +	nt	nt	nt	nt
BkBGTE + AGTE	nt	nt	nt	14.53 ± 0.29 +	nt	nt	nt	nt
BBGTE + BkBGTE	nt	nt	nt	nt	nt	nt	-	-

Note: (nt)—not tested; (-) no inhibition zone; (**++**) synergistic antimicrobial effect; (**+**) basic antimicrobial effect.

**Table 5 antibiotics-13-00181-t005:** The GTE % corresponding to the minimum inhibitory concentration.

Studied Microorganisms	OGTE	AGTE	BMGTE	WGTE	BBGTE	BkBGTE	BCGTE
Gram-positive bacteria							
*B. cereus*	10	10	10	20	20	20	40
*S. aureus*	20	40	20	30	30	60	70
*E. faecalis*	20	40	50	40	30	60	-
*L. monocytogenes*	20	40	30	30	30	60	70
Gram-negative bacteria							
*P. vulgaris*	30	10	20	30	20	10	20
*P. aeruginosa*	40	60	40	40	50	40	50
*E. coli*	50	70	50	60	50	50	70
*S. enterica*	30	60	50	50	40	60	100
Yeast							
*S. cerevisiae*	50	90	40	70	60	60	80

**Table 6 antibiotics-13-00181-t006:** The GTE % corresponding to the minimum bactericidal concentration.

Studied Microorganisms	OGTE	AGTE	BMGTE	WGTE	BBGTE	BkBGTE	BCGTE
Gram-positive bacteria							
*B. cereus*	-	-	-	-	-	-	-
*S. aureus*	60	-	80	-	80	60	-
*E. faecalis*	50	70	70	90	30	70	-
*L. monocytogenes*	70	60	50	80	60	40	-
Gram-negative bacteria							
*P. vulgaris*	-	-	-	-	-	-	-
*P. aeruginosa*	-	100	-	-	70	-	-
*E. coli*	70	-	-	-	-	-	-
*S. enterica*	70	-	100	100	80	-	-
Yeast							
*S. cerevisiae*	90	-	100	-	100	100	-

**Table 7 antibiotics-13-00181-t007:** Antimicrobial features of the GTEs as revealed by different methods on the studied microorganisms.

Extract	Method	Microorganism								
		*B. cereus*	*S. aureus*	*E. faecalis*	*L. monocytogenes*	*P. vulgaris*	*P. aeruginosa*	*E. coli*	*S. enterica*	*S. cerevisiae*
OGTE	ADM	+++	+++	+	+	+++	nd	nd	nd	nd
	MIC	+++	+++	+++	+++	+++	++	++	+++	++
	MBC	nd	++	++	+	nd	nd	+	+	+
	MIC/MBC									
AGTE	ADM	+	nd	+++	nd	+++	nd	nd	nd	++
	MIC	+++	++	++	++	+++	++	+	++	+
	MBC	nd	nd	+	++	nd	+	nd	nd	nd
BMGTE	ADM	++	nd	nd	nd	++	nd	nd	nd	nd
	MIC	+++	+++	++	+++	+++	++	++	++	++
	MBC	nd	+	+	++	nd	nd	nd	+	+
WGTE	ADM	+++	nd	++	+++	+	nd	nd	nd	nd
	MIC	+++	+++	++	+++	+++	++	++	++	++
	MBC	nd	nd	+	+	nd	nd	nd	+	nd
BBGTE	ADM	++	nd	++	nd	+++	nd	nd	nd	+
	MIC	+++	+++	+++	+++	+++	++	++	++	++
	MBC	nd	+	+++	++	nd	+	nd	+	+
BkBGTE	ADM	nd	+++	nd	+++	+	nd	nd	nd	+
	MIC	+++	++	++	++	+++	++	++	++	++
	MBC	nd	++	+	++	nd	nd	nd	nd	+
BCGTE	ADM	nd	nd	nd	+	nd	nd	nd	nd	+
	MIC	+	+	nd	+	+++	++	+	+	+
	MBC	nd	nd	nd	nd	nd	nd	nd	nd	nd

(ADM)—agar diffusion method; (MIC)—minimal inhibition concentration; (MBC)—minimal bactericidal concentration; (+)—reduced but detectable effect on studied microorganisms at high 70–100% GTE concentrations; (++)—medium size effect on the studied microorganisms at 40–60% GTE concentration range; (+++)—significant effect on studied microorganisms for the lowest 10–30% GTE concentration interval; (nd)—not detected (had absolutely no effect on the studied microorganisms even at 100% concentrated GTE).

**Table 8 antibiotics-13-00181-t008:** The main MS transitions of the standards.

Name of Standard	Retention Time, min	*m*/*z*, and Main Transition	MRM
Caffeic acid	13.8	179.0 > 135.0	Negative
Chlorogenic acid	11.9	353.0 > 191.0	Negative
Ferulic acid	18.4	193.0 > 134.0	Negative
Gallic acid	7.0	168.9 > 125.0	Negative
Salicylic acid	23.5	137.0 > 93.0	Negative
Apigenina	28.1	269.0 > 117.0	Negative
Catechin	10.3	289.0 > 202.9	Negative
Chrysin	29.7	253.0 > 143.0	Negative
Hyperoside	20.3	463.1 > 300.0	Negative
Kaempferol	27.9	285.0 > 187.0	Negative
Luteolin	26.8	287.0 > 153.0	Positive
Luteolin-*7*-*O*-glucosid	19.9	447.0 > 284.9	Negative
Naringenin	26.2	271.0 > 119.0	Negative
Quercetin	25.4	300.9 > 151.0	Negative
Rutoside	20.2	609.0 > 300.0	Negative

**Table 9 antibiotics-13-00181-t009:** The standards used for LC/MS analysis.

Name of Standard	Origin	Concentration Range, mg/mL	Calibration Curve Equation	Correlation Factor	Detection Limit, μg/mL	Quantification Limit, μg/mL
Caffeic acid	Phytolab, Vestenbergsgreuth, Germany	0.11–1.10	Area = 4 × 10^7^ × conc[mg/mL] − 319,689	0.9998	3.20	4.80
Chlorogenic acid		0.13–1.30	Area = 2 × 10^8^ × conc[mg/mL] − 269,699	0.9997	5.00	8.00
Ferulic acid		0.100–1.000	Area = 5 × 10^6^ × conc[mg/mL] − 50,000	0.9992	4.00	6.00
Gallic acid		0.107–1.070	Area = 8 × 10^6^ × conc[mg/mL] − 37,131	0.9999	1.90	2.80
Apigenina		0.10–0.98	Area = 2 × 10^8^ × conc[mg/mL] + 15,916	0.9999	0.20	0.30
Hyperoside		0.012–0.107	Area = 4 × 10^8^ × conc[mg/mL] − 567,182	0.9986	0.60	0.90
Kaempferol		0.10–1.00	Area = 10^7^ × conc[mg/mL] − 20,574	0.9996	0.80	1.20
Luteolin		0.01–0.10	Area = 2 × 10^8^ × conc[mg/mL] − 2295.4	0.9977	0.05	0.07
Luteolin-*7-O*-glucosid		0.07–0.70	Area = 1 × 10^9^ × conc[mg/mL] − 700,317	0.9990	3.00	4.00
Naringenin		0.16–1.60	Area = 3 × 10^8^ × conc[mg/mL] − 43,443	0.9999	0.60	0.90
Quercetin		0.09–0.91	Area = 5 × 10^7^ × conc[mg/mL] − 9556	0.9964	0.80	1.10
Rutoside		0.17–1.70	Area = 2 × 10^8^ × conc[mg/mL] − 191,937	0.9996	4.00	6.00
Salicylic acid	Merck, Darmstadt, Germany	0.16–1.60	Area = 4 × 10^7^ × conc[mg/mL] + 44,120	0.9997	1.50	2.00
Catechin		0.10–1.01	Area = 5 × 10^6^ × conc[mg/mL] − 1706	0.9984	1.00	2.00
Chrysin		0.10–1.00	Area = 1 × 10^8^ × conc[mg/mL] − 82,818	0.9997	3.00	5.00

## Data Availability

Data can be made available by the corresponding author on request.

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
