# Peer review of "Specific Antimicrobial Activities Revealed by Comparative Evaluation of Selected Gemmotherapy Extracts"

_antibiotics, 2024, doi:10.3390/antibiotics13020181_

Round 1

Reviewer 1 Report

Comments and Suggestions for Authors

The study was interesting which investigated the antimicrobial of gemmotherapy extracts and mixed of them.

There are some points to be improved for more understandable and for manuscript correction.

1. How substantial and practicable of buds and young shoots of the plants studies? Please discuss in case of upscaling application and linking to the ultimate goal that the author stated in the aim of the research section? 

2. The name of plants, the authors used common name in the text, while used scientific name in the table results. This makes reader difficulty to catch up the content. Please use the common name instead.

3. Table 1 should be present after Table 2 for more understandable. The first column of B. cereus, S. aureus and so on should be Gram-positive in stead of Gram-negative.

4. Please add the novel point to this manuscript.

Author Response

Thank you for the comments. We will address the raised points by making the following modifications:

  1. How substantial and practicable of buds and young shoots of the plants studies are? Please discuss in case of upscaling application and linking to the ultimate goal that the author stated in the aim of the research section.

Our extracts are made based on standard procedures, and the plant material is provided from specific plantations, orchards and forests according to the given species. More importantly, the environmental conditions are also strictly monitored, and because no chemicals are applied in the region where the plant material is produced all the used buds and young shoots are of organic origin. Regarding the upscaling of GTE production, presumably, it is a matter of proper investment and management, but on a strict professional ground, the application of very strict quality control for GTEs is more crucial. The quality control of GTEs implies quantitative and qualitative analytical chemistry. This is the reason why we have included a new table with some quantified polyphenols.  

The ultimate goal of our research is to study the antimicrobial activity of the combination of GTEs as their synergy, addition or inhibition could provide us further hints regarding their combination to augment their associated physiological effects. On the other hand, if we identify a GTE that has an anti-inflammatory effect but also has a bactericidal effect, it would not be wise to use it as an anti-inflammatory agent because interfering with the microbiota could be detrimental. Therefore if an antimicrobial interaction is detected between some GTEs then their possible combination for other putative beneficial physiological effects should be further investigated. This last sentence we have included into the mentioned paragraph.

Also in the Discussion, the introductory paragraphs deal with this consideration.

  1. The name of plants, the authors used common name in the text, while used scientific name in the table results. This makes reader difficulty to catch up the content. Please use the common name instead.

Very good remark. Done.

  1. Table 1 should be present after Table 2 for more understandable. The first column of B. cereus, S. aureus and so on should be Gram-positive instead of Gram-negative.

Corrections have been made.

  1. Please add the novel point to this manuscript.

We have modified the ms by adding a new paragraph to the RESULTS, in which we would describe the quantitation of some but relevant polyphenols for the studied GTEs. The data have been included in Table 1.  

We thank you for the comments because they made us get back to our early concerns regarding GTEs quality assurance. It took us a couple of years to reach the current methodology and these days we are focused on the comparative analyses of GTEs.    

Reviewer 2 Report

Comments and Suggestions for Authors

I am satisfied with the results provided in the manuscript. I recommend accepting the article in its current form.

Author Response

Thank you for the support. We made some corrections and minor changes to the ms as it was suggested. these are shown in yellow.

Reviewer 3 Report

Comments and Suggestions for Authors

The authors have presented antimicrobial data on the gemmotherapy extracts of 7 different plant species. They have used a combination of data from agar diffusion, MIC, MBC and a combination of different plant GTEs to showcase their activity against the common food-derived pathogens. They have discussed the effectiveness of the different GTEs in great details and I appreciate the thoroughness of the work.

I just have a couple of comments:

Section 4.2 (Sample preparation and extraction) is a little confusing to me. I get the solvent extraction, stirring for 20 days and separation of the solvents from the plant extracts. They mentioned the plant material being pressed and combined the solvent extracts. During this pressing do they add more solvent? If you can write it in more details, so that anybody can follow the procedure, that will be more helpful. I also tried finding the European pharmacopeia method, but could not get it. Maybe citing a reference will also be useful.

In Table 6 the color coding was a little confusing. Is there a meaning to the color codes? If so please mention it. I could not find an explanation for the colors. 

Author Response

Thank you for the comments. We will address the raised points by making the following modifications:

  Section 4.2 (Sample preparation and extraction) is a little confusing to me.  

The relevant details were included from the European Pharmacopoeia, monograph 07/2022:2371, method 2.1.3.

In Table 6 the color coding was a little confusing. Is there a meaning to the color codes? If so please mention it. I could not find an explanation for the colors.

 The colors have been removed.

Reviewer 4 Report

Comments and Suggestions for Authors

The research on specific antimicrobial activities of gemmotherapy extracts has significant implications for the development of new antimicrobial treatments and therapies. Here are some potential implications based on the findings of the study:

Exploration of Natural Substances: The study adds to the growing body of evidence supporting the potential of natural substances, such as gemmotherapy extracts, as sources of antimicrobial agents. This suggests that natural substances could serve as valuable reservoirs for the development of novel antimicrobial treatments.

Combatting Antimicrobial Resistance: With the global concern over antimicrobial resistance (AMR), the exploration of alternative antimicrobial agents becomes crucial. The potential antimicrobial activities of gemmotherapy extracts offer a new avenue for addressing AMR and developing treatments effective against drug-resistant microorganisms .

Diversification of Antimicrobial Agents: The limited number of available types of antibiotics has contributed to the rise of drug-resistant bacteria. By exploring the antimicrobial properties of gemmotherapy extracts, the research contributes to diversifying the range of available antimicrobial agents, potentially offering new options for combating microbial infections .

Development of Plant-Based Therapeutics: The interest in natural antimicrobials, including plant extracts, has been increasing. The study adds to this interest by highlighting the antimicrobial potential of gemmotherapy extracts, thereby contributing to the exploration of plant-based therapeutics for microbial infections

Support for Sustainable Healthcare: Natural substances, including plant extracts, offer the potential for sustainable healthcare solutions. By demonstrating the antimicrobial activities of gemmotherapy extracts, the research supports the exploration of sustainable and eco-friendly alternatives for antimicrobial treatments and therapies .

In conclusion, the research on specific antimicrobial activities of gemmotherapy extracts has the potential to contribute to the development of new antimicrobial treatments and therapies by exploring natural substances, diversifying the range of available antimicrobial agents, and supporting sustainable healthcare solutions.

This work presents important elements of the extracts, I suggest it be accepted in this form.

Author Response

Thank you for your support.  some minor modifications were made as the other reviewer suggested